# RiskQ: Risk-sensitive Multi-Agent Reinforcement Learning Value Factorization

**Siqi Shen**[ab], **Chennan Ma**[ab], **Chao Li**[ab], **Weiquan Liu**[ab],
**Yongquan Fu**[c]*, **Songzhu Mei**[c], **Xinwang Liu**[c], **Cheng Wang**[ab]
[a]Fujian Key Laboratory of Sensing and Computing for Smart Cities,
School of Informatics, Xiamen University (XMU), China
[b]Key Laboratory of Multimedia Trusted Perception and Efficient Computing, XMU, China
[c]School of Computer, National University of Defense Technology, China
{siqishen,cwang}@xmu.edu.cn, {yongquanf,xinwangliu}@nudt.edu.cn
{chennanma,chaoli}@stu.xmu.edu.cn

## Abstract

Multi-agent systems are characterized by environmental uncertainty, varying policies of agents, and partial observability, which result in significant risks. In the context of Multi-Agent Reinforcement Learning (MARL), learning coordinated and decentralized policies that are sensitive to risk is challenging. To formulate the coordination requirements in risk-sensitive MARL, we introduce the Risk-sensitive Individual-Global-Max (RIGM) principle as a generalization of the Individual-Global-Max (IGM) and Distributional IGM (DIGM) principles. This principle requires that the collection of risk-sensitive action selections of each agent should be equivalent to the risk-sensitive action selection of the central policy. Current MARL value factorization methods do not satisfy the RIGM principle for common risk metrics such as the Value at Risk (VaR) metric or distorted risk measurements. Therefore, we propose RiskQ to address this limitation, which models the joint return distribution by modeling quantiles of it as weighted quantile mixtures of per-agent return distribution utilities. RiskQ satisfies the RIGM principle for the VaR and distorted risk metrics. We show that RiskQ can obtain promising performance through extensive experiments. The source code of RiskQ is available in https://github.com/xmu-rl-3dv/RiskQ.

## 1 Introduction

In cooperative multi-agent reinforcement learning (MARL) [1], it is important to learn coordinated agent policies to achieve a common goal. However, achieving this goal is challenging due to random rewards, environmental uncertainty, and varying policies among agents. Especially, for scenarios with partial-observability, high-stochastic rewards and state-transitions [2]. In order to efficiently learn MARL policies, many researchers have adopted the centralized training with decentralized execution (CTDE) [3] paradigm, which offers advantages in terms of learning speed and performance. A popular subset of the CTDE methods is the value factorization category [4, 5, 6, 7, 8].

To learn coordinated policies, value factorization algorithms must ensure that the global argmax operator performed on the global state-action value function yields the same outcome as a set of individual argmax operations performed on per-agent utilities. This requirement for coordination is known as the individual-global-max (IGM) principle [6]. The IGM principle takes only the expected return into account, but not the entire distribution of returns that includes potential outcome

---

*Corresponding author

37th Conference on Neural Information Processing Systems (NeurIPS 2023).

events. A method that learns the expected return return may fail in high-stochastic environments with extremely high/low rewards but at low probabilities. For example, users may seek for a big win with low probability in finance or avoid suffering from a huge loss on rare occasions in autonomous driving [9, 10]. Risk refers to the uncertainty of future outcomes in multi-agent systems. By making decisions based on risk, agents can address uncertainty better. Most of the existing MARL value factorization methods do not extensively consider *risk*, which could impact their performance negatively.

Recently, risk-sensitive reinforcement learning (RL) has achieved significant progress in the single agent domain [11, 12]. Instead of optimizing the expectation of return, risk-sensitive RL optimizes a risk measure based on a return distribution. Risk-sensitive value factorization methods should be designed to learn risk-sensitive decentralized policies that are fully consistent with the risk-sensitive centralized counterpart. In order to learn coordinated risk-sensitive policies, the IGM principle needs to be adapted to address cases where expectations are not the only factor. Despite there are a few approaches [13, 14] combing risk-sensitive RL with MARL, how to effectively combine risk-sensitive reinforcement learning with MARL value factorization is still an open question.

In this work, we formulate the coordination requirement in risk-sensitive MARL as *the Risk-sensitive Individual-Global-Max (RIGM) principle*. The principle requires that the optimal joint risk-sensitive action should be equivalent to the collection of each agent's greedy risk-sensitive actions. The RIGM principle is a generalization of the IGM and the distributional IGM (DIGM) principle. Albeit multiple value factorization methods [8, 7, 15] have been proposed to learn policies satisfying the IGM or the DIGM principles, they cannot guarantee the RIGM principle. DRIMA [14] combines risk-sensitive RL with a value factorization method. However, it does not guarantee the RIGM principle for distorted risk measures [9]. RMIX [13] learns risk-sensitive policies which satisfy the RIGM principle only if the risk metric is the expectation operator. It is unclear how to learn coordinated risk-sensitive decentralized policies which satisfy the RIGM principle for general risk measures.

We build, RiskQ, a risk-sensitive MARL value factorization algorithm that satisfies the RIGM principle for risk metrics $\psi_\alpha$, such as VaR (i.e., percentile) and distorted risk measurements [9], where $\alpha$ is a risk parameter. In RiskQ, each agent acts greedily according to a risk value defined as $\psi_\alpha[Z_i]$, where $Z_i$ is a per-agent return distribution utility. RiskQ models the joint return distribution $Z_{jt}$ by combining per-agent return distribution utilities $[Z_i]_{i=1}^n$ with an attention-based mechanism. Specifically, the quantiles $\theta(\omega)$ of the return distribution $Z_{jt}$ is modeled as the weighted sum of the quantiles $\theta_i(\omega)$ of return distribution utilities, where $\omega$ is a quantile sample.

For evaluation, we conduct extensive experiments on risk-sensitive games and the StarCraft II MARL tasks [16]. The experimental results show that RiskQ can obtain promising results in both risk-sensitive and risk-neutral scenarios.

## 2 Background

### 2.1 Dec-POMDPs

We consider cooperative multi-agent reinforcement learning (MARL) scenarios which can be modeled as Decentralized Partially Observable Markov Decision Processes (Dec-POMDPs) [17], represented as tuple $G = \langle \mathcal{S}, \{\mathcal{U}_i\}_{i=1}^N, P, r, \{\mathcal{O}_i\}_{i=1}^N, \{\sigma_i\}_{i=1}^N, N, \gamma \rangle$ for $N$ agents.

$S$ is a finite set of states, and $\mathcal{U}_i$ is the set of discrete actions available to agent $i$. At state $s^t$, a joint action of all agents is defined as $\boldsymbol{u}^t \in \mathcal{U}^N = \mathcal{U}_1 \times \ldots \times \mathcal{U}_N$, with $t$ representing the discrete time step. After performing $\boldsymbol{u}^t$, the environment transitions to a new state $s^{t+1} \in S$, following the transition function $s^{t+1} \sim P(\cdot|s^t, \boldsymbol{u}^t)$, and each agent receives a reward $r^t$ as a result of the state transition. Due to the partial observability of the environment, each agent $i$ receives a local observation $o_i^t \in O_i$, which is drawn from $o_i^t \sim \sigma^i(\cdot|s^t)$. The discounting factor is denoted by $\gamma \in [0, 1)$. Each agent $i$ maintains a local action-observation history $\tau_i = (O_i \times U_i)^*$, where $*$ represents 0 to $T$ ( $T$ denotes the time step). The global action-observation history is denoted as $\tau \in \mathcal{T}^N := \tau_1 \times \ldots \times \tau_N$. Each agent acts according to policy $\pi_i(u_i|\tau_i)$, and the joint policy can be represented as $\pi = <\pi_1, \ldots, \pi_N>$.

## 2.2 Value Function Factorization

For Dec-POMDPs, value function factorization methods learn factorized utility which can be used for the execution of individual agent. The Individual-Global-Max (**IGM**) principle proposed in [6] is important for the realization of value function factorization for MARL. It is defined as follows.

**Definition 1** (IGM). *For a joint state-action value function* $Q_{\text{jt}} : \mathcal{T}^N \times \mathcal{U}^N \mapsto \mathbb{R}$, *where* $\tau \in \mathcal{T}^N$ *is a joint action-observation history and* $\boldsymbol{u}$ *is the joint action, if there exists individual state-action functions* $[Q_i : \mathcal{T}_i \times \mathcal{U}_i \mapsto \mathbb{R}]_{i=1}^N$, *such that the following conditions are satisfied*

$$\arg \max_{\mathbf{u}} Q_{\text{jt}}(\boldsymbol{\tau}, \boldsymbol{u}) = (\arg \max_{u_1} Q_1(\tau_1, u_1), \ldots, \arg \max_{u_n} Q_N(\tau_N, u_N)), \tag{1}$$

*then,* $[Q_i]_{i=1}^N$ *satisfy IGM for* $Q_{\text{jt}}$ *under* $\tau$. *We can state that* $Q_{\text{jt}}(\boldsymbol{\tau}, \boldsymbol{u})$ *is factorized by* $[Q_i(\tau_i, u_i)]_{i=1}^N$.

## 2.3 Distributional RL and Risk

MARL is highly stochastic, and distributional RL could be used to deal with such stochasticity. Distributional RL [18, 19, 20, 21] models the return distribution of state-action pair through $Z(\boldsymbol{\tau}, \boldsymbol{u})$ explicitly. They model full return distribution $Z(\boldsymbol{\tau}, \boldsymbol{u})$ instead of return expectation $Q(\boldsymbol{\tau}, \boldsymbol{u})$. The distribution of return can be approximated through a categorical distribution [18] or a quantile function [19, 20].

The state-action return distribution $Z(\boldsymbol{\tau}, \boldsymbol{u})$ can be modelled using quantile functions $\theta$ of a random variable Z, which is defined as follows.

$$\theta_Z(\boldsymbol{\tau}, \boldsymbol{u}, \omega) = \inf\{z \in \mathcal{R} : \omega \leq CDF_Z(z)\}, \quad \forall \omega \in [0, 1] \tag{2}$$

where $CDF_Z(z)$ is the cumulative distribution function of $Z(\boldsymbol{\tau}, \boldsymbol{u})$. The quantile function $\theta_Z(\omega)$ may be referred as generalized inverse CDF in other literature [22]. For notation simplicity, we denote $\theta_Z(\omega)$ as $\theta(\omega)$.

QR-DQN [19] and IQN [20] model the return function $Z(\tau, u)$ as a mixture of $n$ Dirac functions.

$$Z(\tau, u) = \sum_{i=1}^n p_i(\tau, u, \omega_i) \delta_{\theta(\boldsymbol{\tau}, \boldsymbol{u}, \omega_i)} \tag{3}$$

where $\delta_{\theta(\boldsymbol{\tau}, \boldsymbol{u}, \omega_i)}$ is a Dirac Delta function whose value is $\theta(\boldsymbol{\tau}, \boldsymbol{u}, \omega_i)$. $\omega_i$ is a quantile sample. $p_i(\tau, u, \omega_i)$ is the corresponding probability of $\theta(\boldsymbol{\tau}, \boldsymbol{u}, \omega_i)$. For QR-DQN [19], $p_i(\tau, u, \omega_i)$ can be simplified as $1/n$. For execution, the action with the largest expected return $\arg \max_u \mathbb{E}[Z(\boldsymbol{\tau}, \boldsymbol{u})]$ is chosen.

Analogous to the IGM principle for value function factorization, the Distributional Individual-Global-Max (**DIGM**) principle for value distribution proposed in [22] is defined as follows.

**Definition 2** (DIGM). *Given a set of individual state-action return distribution utilities* $[Z_i(\tau_i, u_i)]_{i=1}^N$ *and a joint state-action return distribution* $Z_{jt}(\boldsymbol{\tau}, \boldsymbol{u})$, *if the following conditions are satisfied*

$$\arg \max_{\mathbf{u}} \mathbb{E}[Z_{jt}(\boldsymbol{\tau}, \boldsymbol{u})] = (\arg \max_{u_1} \mathbb{E}[Z_1(\tau_1, u_1)], \ldots, \arg \max_{u_N} \mathbb{E}[Z_N(\tau_N, u_N)]), \tag{4}$$

*then,* $[Z_i(\tau_i, u_i)]_{i=1}^N$ *satisfy DIGM for* $Z_{jt}$ *under* $\boldsymbol{\tau}$. *We can state that* $Z_{jt}(\boldsymbol{\tau}, \boldsymbol{u})$ *is distributionally factorized by* $[Z_i(\tau_i, u_i)]_{i=1}^N$.

In this work, we use $\psi_\alpha$ to measure the risk from a return distribution Z, where $\alpha$ is the risk level. For example, the Value at Risk metric, $\text{VaR}_\alpha$, estimate the $\alpha$-percentile from a distribution. For $\text{VaR}_\alpha$, a small value for $\alpha$ indicates risk-averse setting, whereas a high value for $\alpha$ means risk-seeking scenarios. Risk-sensitive policies act with a risk measure $\psi_\alpha$.

**Definition 3** (Value at Risk (VaR)). *Value at Risk (VaR) [23] is a popular risk metric which measures risk as the minimal reward might be occur given a confidence level* $\alpha$. *For a random variable Z with cumulative distribution function (CDF), the quantile function* $\theta$ *and a quantile sample* $\alpha \in [0, 1]$, $VaR_\alpha(Z(\boldsymbol{\tau}, \boldsymbol{u})) = \theta(\boldsymbol{\tau}, \boldsymbol{u}, \alpha)$. *This metric is called percentile as well.*

**Definition 4** (Distortion risk measures (DRM)). *Distorted risk measures are weighted expectation of a distribution under a distortion function [9, 10, 24]. The distorted expectation of a random variable Z under g is defined as*

$$\psi(Z) = \int_0^1 g'(\omega)\theta(\omega)d\omega \tag{5}$$

*where $g(\omega) \in [0, 1]$, $g'(\omega)$ is the derivative of $g(\omega)$. There are many distortion functions which can reflect different risk preferences, such as CVaR[25], Wang[26], and CPW[27].*

**Definition 5** (Conditional Value at Risk(CVaR)).

$$CVaR_\alpha(Z) = \mathbb{E}_Z[z|z \le \theta(\alpha)] \tag{6}$$

*where $\alpha$ is the confidence level (risk level), $\theta(\alpha)$ is the quantile function (inverse CDF) defined in (A.1). CVaR is the expectation of values $z$ that are less equal than the $\alpha$-quantile value ($\theta(\alpha)$) of the value distribution. CVaR is a DRM whose $g(\omega) = \min(\omega/\alpha, 1)$.*

Please refer to the Appendix A.1 for the detailed definitions of more distortion risk measures.

## 3 Related Work

### 3.1 Value Factorization

To enable efficient decentralized execution of MARL policies, value factorization approaches are widely adopted in MARL [28, 6, 15]. Most methods focus on the satisfaction of the IGM principle which is important for value factorization. VDN [4] factorizes the value function as the sum of per-agents' utilities. QMIX [5] models monotonic relationships between individual utilities and the value function. QAtten [29] and REFIL [30] use attention mechanisms for value function factorization. QTran [6] transforms the joint state-action value function $Q_{jt}$ into an easy-to-factorize form through linear constraints. QPlex [7] decomposes a value function into a value part and an advantage part. ResQ [8] transforms a value function into the combination of a main and a residual function through masking.

For distributional MARL, the DFAC framework [22] and ResZ [8] satisfy the DIGM principle which is the IGM principle for distributional RL. The DFAC framework factorizes a return distribution through mean-shape decomposition which models the mean and shape of the return distribution separately. ResZ transforms a return distribution into the combination of a main and a residual return distribution. We will show that the DFAC framework and ResZ can not guarantee adherence to the RIGM principle for the VaR risk metric.

### 3.2 Risk-sensitive RL

Many researchers have dedicated themselves to studying risk-sensitive reinforcement learning in single-agent settings [11, 12, 31, 32, 33, 24, 34]. O-RAAC [35] learns a full return distribution for its critic and optimizes the actor's policy according to a risk related metric (such as CVaR). [36] proposes a distributional reinforcement learning algorithm for learning CVaR-optimized policies.

Recently, risk-sensitive reinforcement learning has been adopted in MARL, such as [37, 13]. RMIX [13] combines QMIX and CVaR-optimized agent policies. It learns a value function (*rather than* a return distribution) for each state-action pair, which is further decomposed into per-agent's return distribution utilities. Because the reward for each agent is unknown in cooperative MARL, RMIX learns agents' utilities by using CVaR value as pseudo rewards. RMIX satisfies the RIGM principle only when the risk metric is CVaR and the risk level $\alpha$ is set to 1.

DRIMA [14] separates the sources of risk into cooperation risk and environmental risk. It models joint return distribution as a monotonic mixing of per-agent return distribution utilities. We will show in Sec. 4 that DRIMA does not satisfy the RIGM principle for the CVaR metric.

RiskQ can be used with other MARL-based approaches: communication approaches (COMNet [38], GraphComm [39], DIAL [40], COPA [41]), actor-critic methods (MADDPG [42], MAAC [43]), and other approaches such as MAPPO [44], MASER [45], QRelation [46], ATM [47], and UPDet [48].

## 4 Risk-sensitive Value Factorization

In cooperative MARL, it is crucial to learn decentralized policies consistent with a centralized policy that is conditioned on joint state and joint action. Especially in MARL scenarios with high-stochastic rewards and state transitions, taking risk into consideration is of great importance. However, it is unclear about how to coordinate agents' policies with the consideration of risk.

Key to our approach is the insight that to coordinate agents with risk consideration is important to learn risk-sensitive decentralised policies that are fully consistent with the risk-sensitive centralised counterpart. To ensure this consistency, MARL algorithms only need to ensure that a joint risk-sensitive argmax operation, when performed on the joint state-action value function, yields the same outcome as a collection of individual risk-sensitive argmax operations conducted on per-agent utilities. This insight leads to the following definition.

## 4.1 Risk-sensitive Individual-Global-Max (RIGM) Principle

**Definition 6** (RIGM). *Given a risk metric $\psi_\alpha$, a set of individual return distribution utilities $[Z_i(\tau_i, u_i)]_{i=1}^N$, and a joint state-action return distribution $Z_{jt}(\boldsymbol{\tau}, \boldsymbol{u})$, if the following conditions are satisfied:*

$$\arg\max_{\mathbf{u}} \psi_\alpha[Z_{jt}(\boldsymbol{\tau}, \boldsymbol{u})] = (\arg\max_{u_1}[\psi_\alpha[Z_1(\tau_1, u_1)], \ \ldots, \ \arg\max_{u_N}[\psi_\alpha[Z_N(\tau_N, u_N)]]), \quad (7)$$

*where $\psi_\alpha : Z \times R \to R$ is a risk metric such as the VaR or a distorted risk measure, $\alpha$ is its risk level. Then, $[Z_i(\tau_i, u_i)]_{i=1}^N$ satisfy the RIGM principle with risk metric $\psi_\alpha$ for $Z_{jt}$ under under $\tau$. We can state that $Z_{jt}(\boldsymbol{\tau}, \boldsymbol{u})$ can be distributionally factorized by $[Z_i(\tau_i, u_i)]_{i=1}^N$ with risk metric $\psi_\alpha$.*

The RIGM principle is a generalization of the DIGM and the IGM principle. The DIGM principle is a special case of the RIGM theorem for $\psi = \text{CVaR}$ and $\alpha = 1$ (the expectation operator $\mathbb{E}$ is equal to $\text{CVaR}_1$). If $Z_i$ is a single Dirac Delta Distribution, then the return distribution $Z_i$ becomes a single value(i.e., $Q_i$), and in this case, the IGM principle is equivalent to the RIGM principle when $\psi = \text{CVaR}$ and $\alpha = 1$.

In the following, we discuss whether existing risk-neutral value factorization methods can be simply modified to satisfy the RIGM principle for $\psi_\alpha$, and then discuss limitations of existing risk-sensitive value factorization methods. There are many value factorization methods satisfying the IGM principle. We show in Theorem 1 that simply replacing $Q_i$ with $Z_i$ is insufficient to guarantee that $[Z_i]_{i=1}^N$ satisfy RIGM with risk metric $\psi_\alpha$.

**Theorem 1.** *Given a deterministic joint state-action value function $Q_{jt}$, a joint state-action return distribution $Z_{jt}$, and a factorization function $\Phi$ for deterministic utilities:*

$$Q_{jt}(\tau, u) = \Phi(Q_1(\tau_1, u_1), ..., Q_N(\tau_N, u_N)) \quad (8)$$

*such that $[Q_i]_{i=1}^N$ satisfy IGM for $Q_{jt}$ under $\tau$, the following risk-sensitive distributional factorization:*

$$Z_{jt}(\tau, u) = \Phi(Z_1(\tau_1, u_1), ..., Z_N(\tau_N, u_N)) \quad (9)$$

*is insufficient to guarantee that $[Z_i]_{i=1}^N$ satisfy RIGM for $Z_{jt}(\tau, u)$ with risk metric $\psi_\alpha$.*

We show that factorization methods satisfying the DIGM principle cannot guarantee the satisfaction of the RIGM theorem for the VaR metric.

**Theorem 2.** *Given a joint state-action return distribution $Z_{jt}$, and a distributional factorization function $\Phi$ for the return distribution utilities $[Z_i]_{i=1}^N$ which satisfy the DIGM theorem, the following risk-sensitive distributional factorization:*

$$Z_{jt}(\tau, u) = \Phi(Z_1(\tau_1, u_1), ..., Z_N(\tau_N, u_N)) \quad (10)$$

*is insufficient to guarantee that $[Z_i]_{i=1}^N$ satisfy the RIGM principle for $Z_{jt}(\tau, u)$ with risk metric $\psi_\alpha$.*

Recently, DRIMA [14] and RMIX [13] combine risk-sensitive RL with MARL. Albeit they have demonstrated promising results, they do not explicitly consider the risk-sensitive coordination requirement.

**Theorem 3.** *DRIMA [14] does not guarantee adherence to the RIGM principle for CVaR metric.*

The value function $Q_{jt}(\boldsymbol{\tau}, \boldsymbol{u})$, learned by RMIX [13], can be written as $Q_{jt}(\boldsymbol{\tau}, \boldsymbol{u}) = Q_{mix}(\psi_\alpha[Z_1(\tau_1, u_1)], ..., \psi_\alpha[Z_N(\tau_N, u_N)])$, where $Z_i(\tau_i, u_i)$ represents per-agent return distribution utilities, and $Q_{mix}$ is a monotonically increasing function with respect to $\psi_\alpha[Z_i(\tau_i, u_i)]$. Although it seems that it satisfies the RIGM principle for any risk metric $\psi_\alpha$, its learning algorithm seeks to find the optimal actions that $\arg\max_{\mathbf{u}}[Q_{jt}(\boldsymbol{\tau}, \boldsymbol{u})]$ rather than $\arg\max_{\mathbf{u}} \psi_\alpha[Z_{jt}(\boldsymbol{\tau}, \boldsymbol{u})]$. In essence, RMIX seeks to make sure that $\arg\max_{\mathbf{u}}[Q_{jt}(\boldsymbol{\tau}, \boldsymbol{u})]$ equal to

$(\arg\max_{u_1}[\psi_\alpha[Z_1(\tau_1, u_1)], \ldots, \arg\max_{u_N}[\psi_\alpha[Z_N(\tau_N, u_N)]])$. By this way, RMIX satisfies the RIGM principle only if $\psi_\alpha = \text{CVaR}_1$. Moreover, $Q_{jt}(\boldsymbol{\tau}, \boldsymbol{u})$ is only a value function but not a return distribution.

Please refer to Appendix B.1 for detailed proofs of Theorem 1, 2 and 3. We have discussed the limitations of existing value factorization methods. It becomes apparent that the development of a novel factorization method is needed, specifically one capable of effectively coordinating agents in risk-sensitive scenarios.

## 4.2 RiskQ

In MARL, it is typical to model the factorization function $f$ as either the sum of per-agent utilities or a monotonic increasing function with respect to per-agent utilities. However, risk metrics are generally non-additive, except for variables which are highly dependent on each other (the comonotonicity property) [49, 10]. For instance, let's consider the $\text{VaR}_{0.5}$ metric, the median of a distribution, and $f = \sum Z_i$. It generally holds that $\text{VaR}_{0.5}[Z_1 + Z_2] \neq \text{VaR}_{0.5}[Z_1] + \text{VaR}_{0.5}[Z_2]$. This is due to the fact that the median of the sum of two random variables does not equate to the sum of their medians. The non-additive property of risk metrics makes the explicit modeling of $f$ challenging, as we cannot model $f$ as the sum or the monotonic mixing of per-agents' utilities.

The key to our insights is that instead of modeling the return distribution $Z_{jt}$ using the form of $f(Z_1, ...Z_n)$ explicitly which is a common practice in MARL [4, 5, 8], we can model $Z_{jt}$ implicitly through modeling it using its quantiles. We model the relationships among quantiles of the joint return distribution and the quantiles of per-agent return distribution utilities.

RiskQ utilizes a common distributional RL technique [19], where the return distribution $Z_{jt}(\boldsymbol{\tau}, \boldsymbol{u})$ is represented by a combination of Dirac delta functions $\delta_{\theta(\omega)}$ and the positions $\theta(\omega)$ of the Diracs that are determined through quantile regression. Figure 1 depicts the mixing process of the quantiles of per-agent utilities. For a quantile sample (cumulative probability) $\omega$, the quantile value $\theta(\boldsymbol{\tau}, \boldsymbol{u}, \omega)$ of $Z_{jt}(\boldsymbol{\tau}, \boldsymbol{u})$ is represented as the weighted sum of the quantile value $\sum_{i=1}^{N} k_i \theta_i(\tau, u_i, \omega)$ of return distribution utilities.

We demonstrate through the following theorem that $Z_{jt}$ and $[Z_i]_{i=1}^{N}$ satisfy the RIGM principle for both the VaR risk metric and distorted risk measures (DRM).

**Theorem 4.** *A joint state-action return distribution*

$$Z_{jt}(\boldsymbol{\tau}, \boldsymbol{u}) = \sum_{j=1}^{J} p_j(\boldsymbol{\tau}, \boldsymbol{u}, \omega_j)\delta_{\theta(\boldsymbol{\tau}, \boldsymbol{u}, \omega_j)} \tag{11}$$

$$\theta(\boldsymbol{\tau}, \boldsymbol{u}, \omega_j) = \sum_{i=1}^{N} k_i \theta_i(\tau_i, u_i, \omega_j) \tag{12}$$

*is distributional factorized by $[Z_i(\tau_i, u_i)]_{i=1}^{N}$ with risk metric $\psi_\alpha$, where $J$ is the number of Dirac Delta functions, $\delta_{\theta(\boldsymbol{\tau}, \boldsymbol{u}, \omega_j)}$ is a Dirac Delta function at $\theta(\boldsymbol{\tau}, \boldsymbol{u}, \omega_j) \in \mathbb{R}$, $\theta(\boldsymbol{\tau}, \boldsymbol{u}, \omega_j)$ is a quantile function with sample $\omega_j$, $p_j(\boldsymbol{\tau}, \boldsymbol{u}, \omega_j)$ is the corresponding probability for $\delta_{\theta(\boldsymbol{\tau}, \boldsymbol{u}, \omega_j)}$ of the estimated return distribution $Z_{jt}$, $\theta_i(\tau_i, u_i, \omega)$ is the quantile function (with quantile sample $\omega$) for the return distribution utility $Z_i(\tau_i, u_i)$ of agent $i$ and $k_i \geq 0$.*

We have shown that by modeling the quantiles of $Z_{jt}$ as a weighted sum of quantiles of $[Z_i]_{i=1}^{N}$. $[Z_i]_{i=1}^{N}$ satisfied the RIGM theorem for VaR and DRMs such as CVaR, Wang, and CPW. Detailed proofs supporting this theorem can be found in Appendix B.2.

## 4.3 Neural Networks and Loss

We model the return distribution utility for each agent by a simple neural network. It takes the observation history $\tau_i$ and action $u_i$ as input, then passes them through a MLP, a GRU, and a MLP, and outputs $Z_i(\tau_i, u_i, \omega)$ for each quantile sample $\omega$. For execution, the action which maximizes $\psi_\alpha[Z_i(\tau_i, u_i)]$ is chosen.

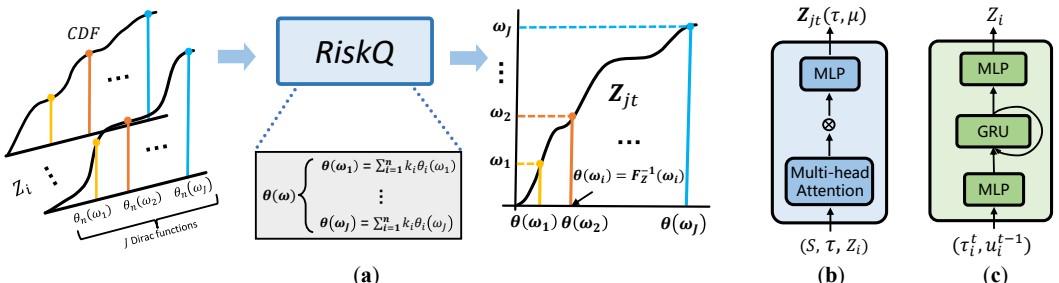

Figure 1: RiskQ overview: (a) quantiles mixing for $Z_{jt}$, (b) mixer function, (c) agent return distribution utility

The mixer function mixes all the quantile $\theta$ of return distribution utility $[Z_i]_{i=1}^N$ into $\theta(\boldsymbol{\tau}, \boldsymbol{u}, \omega) = \sum_{i=1}^N k_i \theta_i(\tau_i, u_i, \omega)$. It takes $[\theta_i(\tau_i, u_i, \omega)]_{i=1}^N$, the state $s$, the observation history $\boldsymbol{\tau}$ as input, and outputs $\theta(\boldsymbol{\tau}, \boldsymbol{u}, \omega)$ for each quantile sample $\omega$. $k_i$ is modelled using a multi-head attention mechanism.

We adopt the Quantile Regression (QR) loss [19] to update the value distribution $Z_{jt}(\boldsymbol{\tau}^k, \boldsymbol{u}^k, \sigma)$. More concretely, QR aims to estimate the quantiles of the return distribution by minimizing the distance between $Z_{jt}(\boldsymbol{\tau}^k, \boldsymbol{u}^k, \sigma)$ and its target distribution $y^k(\boldsymbol{\tau}^k, \boldsymbol{u}^k, \sigma) \triangleq r + \gamma Z_{jt}(\boldsymbol{\tau}^{k+1}, \widetilde{\boldsymbol{u}}, \sigma^-)$. $\widetilde{\boldsymbol{u}} = [\widetilde{u}_i]_{i=1}^N, \widetilde{u}_i = \arg\max_{u_i} \psi_\alpha[Z_i(\tau_i^{k+1}, u_i)]$. $\sigma$ are the parameters of the network, and $\sigma^-$ are the parameters of the target network.

## 5 Evaluation

We study the performance of RiskQ on risk-sensitive games (Multi-agent cliff and Car following games), the StarCraft II Multi-Agent Challenge benchmark (SMAC) [16]. RiskQ can obtain promising performance for risk-sensitive and risk-neutral scenarios. Ablation studies reveal the importance of adhering to the RIGM principle for achieving good performance. Additionally, we have examined the impact of functional representations, risk metrics and risk levels.

### 5.1 Experimental Setup

We select three categories of MARL value factorization methods for comparison: (i) Expected value factorization methods: QMIX [5], QTran [6], QPlex [7], CW QMIX [15], ResQ [8]; (ii) Risk-neutral return distribution (stochastic value) factorization methods: DMIX [22] and ResZ [8]; (iii) Risk-sensitive return distribution factorization methods: RMIX [13] and DRIMA [14]. For robustness, each experiment is conducted at least 5 times with different random seeds. In general, the configuration of RiskQ follows the setup of Weighted QMIX and ResQ. By default, the risk metric used in RiskQ is Wang$_{0.75}$, indicating a risk-averse preference. Please refer to Appendix C for detailed experimental setup and more experimental results.

### 5.2 Multi-Agent Cliff Navigation

In Multi-Agent Cliff Navigation (MACN), introduced in [13], two agents must navigate in a grid-like world to reach the goal without falling into cliff. The agent receives a -1 reward at each time step. If any agent reaches the goal individually, a -0.5 reward will be given to the agents. They will receive $0$ reward if they reach the goal together. An episode is finished once the goal is reached by two agents together or any agent falls into cliff (reward -100). We depict the test return of each learning algorithm in two MACN scenarios: $4 \times 11$ grid map and $4 \times 15$ grid map. As illustrated in Figure 2 (a) and (b), RiskQ achieves the optimal performance in both scenarios, outperforming the two risk-sensitive methods: RMIX and DRIMA. This indicates that learning policies which satisfy the RIGM principle could lead to promising results.

### 5.3 Multi-Agent Car Following game

We design Multi-Agent Car Following (MACF) Game, which is adapted from the single agent risk-sensitive environment [35]. In MACF, there are two agents, each controlling one car, with the

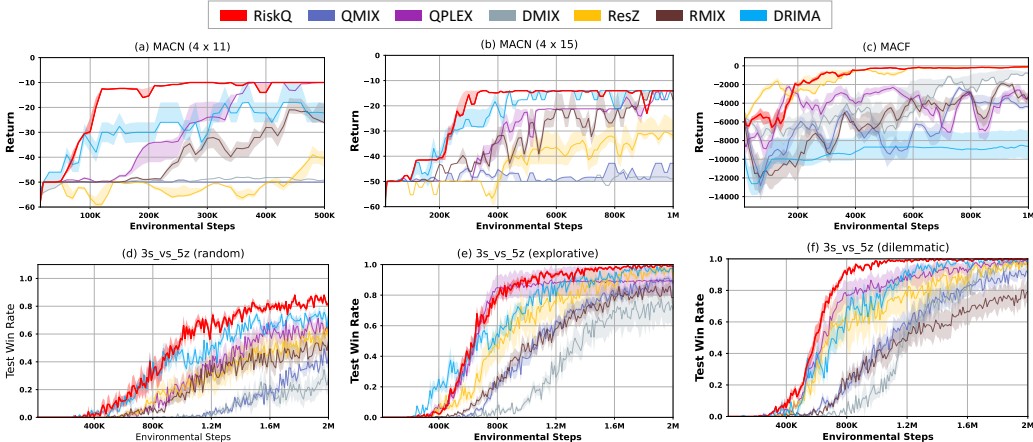

Figure 2: The return of the MACN (a-b) and the MACF (c) environment; the test win rate of random (d), explorative (e) and dilemmatic (f) 3s_vs_5z scenario of SMAC.

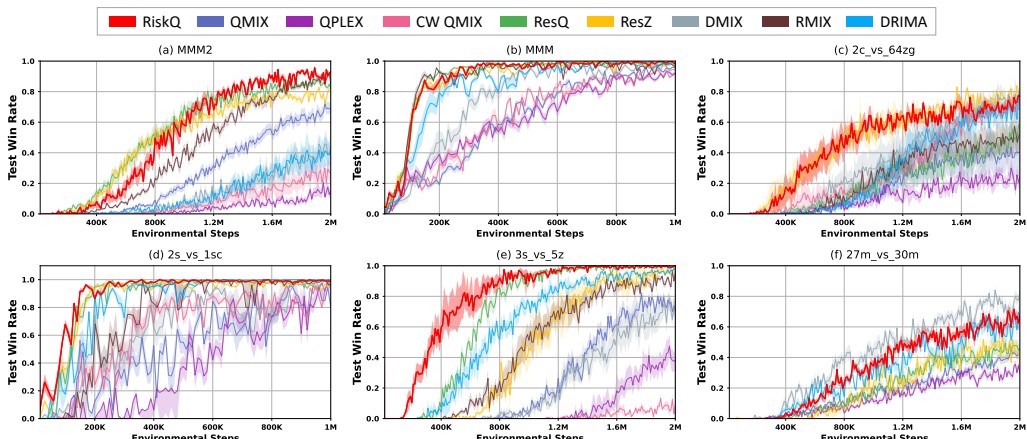

Figure 3: Win Rate of the StarCraft Multi-Agent Scenarios.

task of one car following the other to reach a goal. Each car can observe the current position and speed of other cars within its observation range. The agent has a fixed action space which determines its acceleration. At each time step, agent will receive a negative reward. And the cars will *crash* with some probability if their speed exceed a speed-threshold, and a negative reward is given to the agents. To adapt the game to cooperative MARL, agents move within each other's observation will receive a positive reward. Once the agents reach the goal together, a big reward is given to them and the episode is terminated. As shown in Figure 2 (c), RiskQ exhibits superior performance to both risk-neutral and risk-sensitive algorithms. Moreover, in order to verify that this performance improvement is due to risk considerations, we also study the number of crashes. For this value, as it is shown in the appendix (Figure 7), albeit RiskQ does not optimize for it, RiskQ achieves zero crash with the fastest learning rate.

## 5.4 StarCraft II

SMAC is a well-known benchmark which comprises two teams of agents engaging in combat scenarios. Following the evaluation protocol of DRIMA [14] for risk-averse SMAC, we first study the performance of RiskQ in explorative, dilemmatic and random settings for the 3s_vs_5z scenario. In the explorative setting, agents behave heavily exploratory during training, thus they must consider the risk brought on by heavily exploration of other agents. In the dilemmatic setting, agents have increased exploratory behaviors and they are punished by their decreased heath. In this setting, learning algorithms should consider risk to prevent the learning of locally optimal policies. In the random setting, one agent performs random actions 50% of the time during testing. As depicted in Figure 2 (d-f), RiskQ obtains the best performance. Please refer to Figure 9 in Appendix C.4 for more

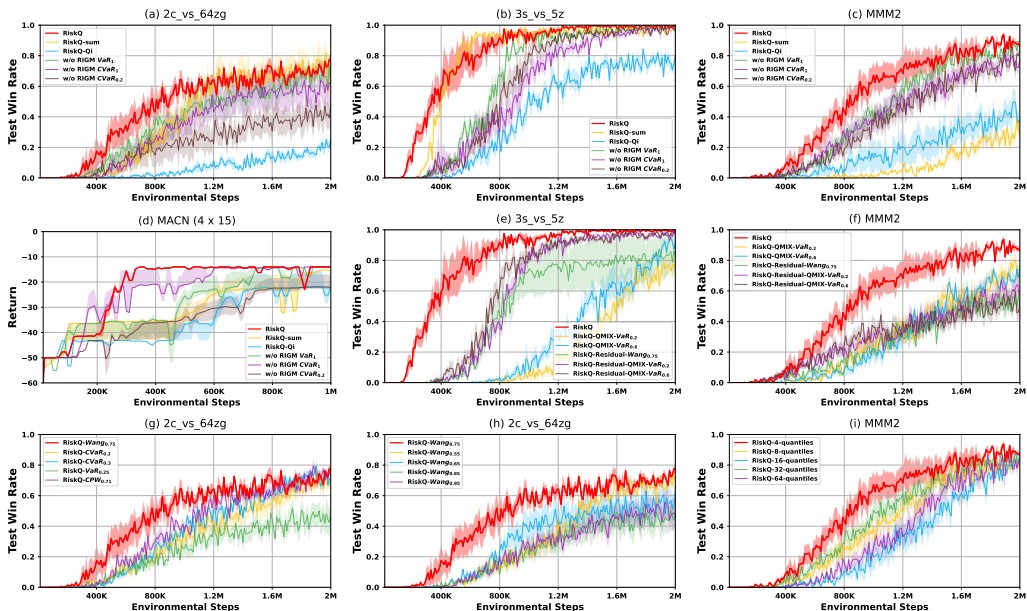

Figure 4: Ablation Study: (a-d) impact of not satisfying RIGM and different design of RiskQ mixer; (e-f) impact of functional representation limitations; (g-i) impact of different risk metrics, risk levels, and number of quantiles

results in these risk-sensitive SMAC scenarios. Combining previous results from the MACN and the MACF environments, we can conclude that RiskQ can yield promising results in environments that require risk-sensitive cooperation.

Then we evaluate the performance of RiskQ in the standard SMAC setting. As demonstrated in Figure 3 (a) and (b), RiskQ achieves the best performance in the MMM2 scenarios. In MMM2, RMIX achieves the second best results in the end. This demonstrates that it is important to consider risk in highly stochastic environments. RiskQ is among the best performing algorithms in the MMM scenario. Notably, RiskQ satisfies the RIGM principle for DRMs, further suggesting the necessity of coordinated risk-sensitive cooperation.

The test win rates for the 2c_vs_64zg and 2s_vs_1sc scenarios are shown in Figure 3 (c) and (d). For the 2c_vs_64zg scenario, RiskQ and ResZ are the best performing algorithms, while RMIX performs poorly in this scenario. Albeit DRIMA matches the performance of RiskQ in the end, its learning speed is much slower than that of RiskQ. For the 2s_vs_1sc scenario, RiskQ, ResQ and ResZ achieve optimal performance, with RiskQ achieving near-optimal performance merely after 0.3 million steps. Furthermore, DRIMA learns slower than RiskQ, and the performance of RMIX is unstable.

For the 3s_vs_5z scenario, as illustrated in Figure 3 (e), RiskQ achieves the optimal performance after only 1.2 million training steps. The risk-sensitive algorithms DRIMA and RMIX do not match up to the performance of RiskQ. As for the 27m_vs_30m scenario, RiskQ is the second-best algorithm.

## 5.5 Ablation Study and Discussion

To investigate the reasons behind RiskQ's promising results, we analyze different designs of RiskQ on the SMAC and the MACN scenarios. First, we study the necessity of satisfying the RIGM principle by making about 50% of RiskQ agents follow different risk measures. In Figure 4 (a-d), w/o RIGM $\text{VaR}_1$, w/o RIGM $\text{CVaR}_1$ and w/o RIGM $\text{CVaR}_{0.2}$ indicate that about 50% of agents act according to the $\text{VaR}_1$ (risk-seeking), $\text{CVaR}_1$ (risk-neutral) and the $\text{CVaR}_{0.2}$ (risk-averse) metrics, respectively. These risk measures are not the risk measure $\psi_\alpha = \text{Wang}_{0.75}$ used by other agents. As depicted in Figure 4 (a-d), RiskQ performs poorly in all the three cases, highlighting the importance of satisfying the RIGM principle that agents act according to the same risk measure.

Moreover, we analyze different designs of the RiskQ mixer through two variants: RiskQ-Sum and RiskQ-Qi. Instead of using the attention mechanism, RiskQ-Sum models the percentile $\theta(\boldsymbol{\tau}, \boldsymbol{u}, \omega)$

as the sum of the percentiles of per-agent's utilities $\sum_{i=1}^{N} \theta_i(\tau_i, u_i, \omega)$. RiskQ-Qi represents each percentile $\theta(\boldsymbol{\tau}, \boldsymbol{u}, \omega)$ as the expectation of state-action function. As shown in Figure 4 (a-d), both two variants perform inferior to RiskQ in most cases.

By modeling the quantiles $\theta(\omega)$ of joint return distribution as the weighted sum of the quantiles $\theta_i(\omega)$ of each agent's return distribution, RiskQ suffers from representation limitations. To study whether the representation limitations impact the performance of RiskQ, we design three variants of RiskQ: RiskQ-QMIX, RiskQ-Residual, and RiskQ-Residual-QMIX. RiskQ-QMIX models the monotonic relations among $\theta_i(\omega)$ and $\theta(\omega)$ in the manner of QMIX [5]. RiskQ-Residual models the joint value distribution without representation limitations by using residual functions [8]. RiskQ-Residual-QMIX combines RiskQ-Residual and QMIX. RiskQ-QMIX and RiskQ-Residual-QMIX satisfy the RIGM principle for the VaR metric, and RiskQ-Residual satisfies the RIGM for the VaR and DRM metrics. Please refer to the appendix B.3, B.4 and B.5 for further details and their proofs.

We evaluate the performance of the three RiskQ variants using three different risk metrics (Wang$_{0.75}$, VaR$_{0.2}$, and VaR$_{0.6}$). In Figure 4 (e) and (f), a method with its risk metric is denoted as method-metric. For example, RiskQ-QMIX-VaR$_{0.2}$ represents RiskQ-QMIX with the risk metric VaR$_{0.2}$. As can be observed from Figure 4 (e) and (f), although these three variants can model more complex functional relationships among quantiles, their performance is unsatisfactory for the 3s_vs_5z and MMM2 scenarios. This suggests that the representation limitation of RiskQ does not significantly impact its performance. Finding a better network architecture to make the algorithm free from representation limitations and have better performance is a prospective future work.

To systematically evaluate RiskQ, we evaluate the impact of various risk metrics, risk levels and number of percentiles. The respective results are depicted in in Figure 4 (g-i). The variations in these factors could impact the performance of RiskQ.

RiskQ uses QR-DQN [19] and IQN [20] to learn value distributions. It shares QR-DQN and IQN's converging property. As stated in [50] and [51], the greedy distributional Bellman update operator of IQN is not a contraction mapping, which is an inherent drawback of the distributional RL. Recently, [52] modified IQN with a new distributional Bellman operator, indicating the optimal CVaR policy corresponds to a fixed point. However, but its general convergence is unclear. As shown in Appendix (Figure 19), the new method that combines RiskQ with [52] performs poorly. This suggests that remedying the non-contraction mapping issues may not be important enough for performance improvement as existing risk-sensitive RL methods (e.g., IQN) are already working well.

For risk-sensitive exploration, we have combined RiskQ with LQN [53], the results are depicted in the Appendix (Figure 20). It shows that risk-sensitive exploration is a new direction of the future work.

# 6   Conclusion

It is important to coordinate behaviors of multi-agents in risk-sensitive environments. We have formulated the coordination requirement as the risk-sensitive individual-global-maximization (RIGM) principle which is a generalization of the individual-global-maximization (IGM) principle and the Distributional IGM principle. Existing multi-agent value factorization does not satisfy the RIGM principle for common risk metrics such as the Value at Risk (VaR) or distorted risk measures. We propose, RiskQ, a risk-sensitive value factorization approach for Multi-Agent Reinforcement Learning (MARL). RiskQ satisfies the RIGM theorem for the VaR and distorted risk measures via modeling the quantile of joint state-action return distribution as weighted sum of the quantiles of per-agent return distribution utilities. We show that RiskQ can obtain promising results through extensive experiments.

**Acknowledgement**   This work was partially supported by the National Natural Science Foundation of China (No. 61972409), by the FuXiaQuan National Independent Innovation Demonstration Zone Collaborative Innovation Platform (No.3502ZCQXT2021003), the Fundamental Research Funds for the Central Universities (No. 20720230033), by PDL (2022-PDL-12), by the China Postdoctoral Science Foundation (No.2021M690094). We would like thank the anonymous reviewers for their valuable suggestions.

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

# Appendix

## A  Background

### A.1  Distributional RL and Risk

In this work, we **interchangeably** use the terms: stochastic value function and return distribution. The state-action return distribution $Z(\boldsymbol{\tau}, \boldsymbol{u})$ can be modelled using quantile functions $\theta$ of a random variable Z, which is defined as follows.

$$\theta_Z(\boldsymbol{\tau}, \boldsymbol{u}, \omega) = \inf\{z \in \mathcal{R} : \omega \leq CDF_Z(z)\}, \quad \forall \omega \in [0, 1] \tag{A.1}$$

where $CDF_Z(z)$ is the cumulative distribution function of $Z(\boldsymbol{\tau}, \boldsymbol{u})$. The quantile function $\theta_Z(\omega)$ may be referred as generalized inverse CDF in other literature [22]. For notation simplicity, we denote $\theta_Z(\omega)$ as $\theta(\omega)$.

QR-DQN [19] and IQN [20] model the stochastic value function $Z(\tau, u)$ as a mixture of $n$ Dirac functions.

$$Z(\tau, u) = \sum_{i=1}^{n} p_i(\tau, u, \omega_i)\delta_{\theta(\boldsymbol{\tau}, \boldsymbol{u}, \omega_i)} \tag{A.2}$$

where $\delta_{\theta(\boldsymbol{\tau}, \boldsymbol{u}, \omega_i)}$ is a Dirac Delta function whose value is $\theta(\boldsymbol{\tau}, \boldsymbol{u}, \omega_i)$. $\omega_i$ is a quantile sample. $p_i(\tau, u, \omega_i)$ is the corresponding probability of $\theta(\boldsymbol{\tau}, \boldsymbol{u}, \omega_i)$. For QR-DQN [19], $p_i(\tau, u, \omega_i)$ can be simplified as $1/n$. For execution, the action with the largest expected return $\arg\max_u \mathbb{E}[Z(\boldsymbol{\tau}, \boldsymbol{u})]$ is chosen.

In this work, we use $\psi_\alpha$ to measure the risk from a return distribution $Z$, where $\alpha$ is the risk level. For example, the Value at Risk metric, $\text{VaR}_\alpha$, estimate the $\alpha$-percentile from a distribution. For $\text{VaR}_\alpha$, a small value for $\alpha$ indicates risk-averse setting, whereas a high value for $\alpha$ means risk-seeking scenarios. Risk-sensitive policies act with a risk measure $\psi_\alpha$. In this work, we interchangeably use the terms: stochastic value function and return distribution.

**Definition 7** (Value at Risk (VaR)). *Value at Risk (VaR) [23] is a popular risk metric which measures risk as the minimal reward might be occur given a confidence level $\alpha$. For a random variable Z with cumulative distribution function (CDF), the quantile function $\theta$ and a quantile sample $\alpha \in [0, 1]$, $VaR_\alpha(Z(\boldsymbol{\tau}, \boldsymbol{u})) = \theta(\boldsymbol{\tau}, \boldsymbol{u}, \alpha)$. This metric is called percentile as well.*

**Definition 8** (Distortion risk measure (DRM)). *Distorted expectation metrics [9, 10, 24], such as CVaR [25], CPW [27], and Wang [26], are weighted expectation of return distribution under a distortion function [9, 10, 24]. The distorted expectation of a random variable Z under g is defined as*

$$\psi(Z) = \int_0^1 g'(\omega)\theta(\omega)d\omega \tag{A.3}$$

*where $g(\omega) \in [0, 1]$, $g'(\omega)$ is the derivative of $g(\omega)$.*

**Definition 9** (Conditional Value at Risk(CVaR)).

$$CVaR_\alpha(Z) = \mathbb{E}_Z[z|z \leq \theta(\alpha)] \tag{A.4}$$

*where $\alpha$ is the confidence level (risk level), $\theta(\alpha)$ is the quantile function (inverse CDF) defined in (A.1). CVaR is the expectation of values z that are less equal than the $\alpha$-quantile value ($\theta(\alpha)$) of the value distribution. CVaR is a DRM whose $g(\omega) = \min(\omega/\alpha, 1)$.*

**Definition 10** (Wang). *Wang is a DRM proposed in [26]. Its $g(\omega) = \Phi(\Phi^{-1}(\omega) + \alpha)$, where $\Phi$ is the CDF of the Gaussian distribution. The Wang measure is risk-averse if $\alpha > 0$ or risk-seeking for $\alpha < 0$.*

**Definition 11** (CPW). *CPW is a DRM proposed in [27]. Its $g(\omega) = \omega^\alpha/(\omega^\alpha + (1 - \omega)^\alpha)^{\frac{1}{\alpha}}$. Researchers found that $\alpha = 0.71$ matches human decision preference.*

**Definition 12.** *Risk-sensitive greedy policy[20] for a value distribution $Z(s, u)$ with a risk measure $\psi_\alpha$ is defined as*

$$\pi_{\psi_\alpha}(s) = \arg\max_u \psi_\alpha[Z(s, u)] \tag{A.5}$$

In this work, we assume the argmax operator is unique, the action with smallest index is selected to break ties if a tie exists.

# B RiskQ Theorems and Proofs

In this section, we show that methods satisfying the IGM principle are insufficient to guarantee the RIGM principle for risk metrics such as the VaR and DRM metrics in Theorem 1. And we show that methods satisfying the DIGM principle are insufficient to guarantee the RIGM principle for the VaR or DRM metrics in Theorem 2. Further, we show that the risk-sensitive algorithm DRIMA does not satisfy the RIGM principle for the VaR metric in Theorem 3.

## B.1 Methods satisfying IGM or DIGM principle

Simply replacing $Q_i$ with $\psi_\alpha(Z_i)$ is insufficient to guarantee $[Z_i]_{i=1}^n$ satisfy RIGM for VaR and distorted expectation metrics.

**Theorem 1.** *Given a deterministic joint action-value function $Q_{jt}$, a stochastic joint action-value function $Z_{jt}$, and a factorization function $\Phi$ for deterministic utilities:*

$$Q_{jt}(\tau, u) = \Phi(Q_1(\tau_1, u_1), ..., Q_n(\tau_n, u_n)) \tag{B.6}$$

*such that $[Q_i]_{i=1}^n$ satisfy IGM for $Q_{jt}$ under $\tau$, the following risk-sensitive distributional factorization:*

$$Z_{jt}(\tau, u) = \Phi(Z_1(\tau_1, u_1), ..., Z_n(\tau_n, u_n)) \tag{B.7}$$

*is insufficient to guarantee that $[Z_i]_{i=1}^n$ satisfy RIGM for $Z_{jt}(\tau, u)$ with risk metric $\psi_\alpha$ such as the VaR metric and the distorted risk measures.*

*Proof.* We first show that VDN does not guarantee the RIGM for the VaR metric, then we show that QMIX does not guarantee the RIGM principle for the CVaR metric (a distorted risk measures). We prove this theorem by contradiction.

The VDN [4] algorithm is factorization methods that satisfy the IGM theorem but it cannot guarantee the RIGM principle. VDN model $Q_{jt}(\boldsymbol{\tau}, \boldsymbol{u}) = \sum_{i=1}^n Q_i(\tau_i, u_i)$. Simply replacing the utility $Q_i$ as $Z_i$. The function becomes $Z_{jt}(\boldsymbol{\tau}, \boldsymbol{u}) = \sum_{i=1}^n Z_i(\tau_i, u_i)$.

We consider a degenerated case where there are two agents and single full observable state $s$. Agents have two actions: $a$ and $b$. The probability distribution function for $Z_i(\tau_i, a)$ and $Z_i(\tau_i, b)$ is defined as follows.

$$p(Z_i(\tau_i, a)) = \begin{cases} 50\% & Z_i = 0.25 \\ 50\% & Z_i = 1 \end{cases} \tag{B.8}$$

$$p(Z_i(\tau_i, b)) = \begin{cases} 50\% & Z_i = 0 \\ 50\% & Z_i = 100 \end{cases} \tag{B.9}$$

We assume that $Z_2(\tau_2, u_2) = Z_1(\tau_1, u_1)$. For VDN, $Z_{jt}(\boldsymbol{\tau}, \boldsymbol{u}) = Z_1(\tau_1, u_1) + Z_2(\tau_2, u_2)$. The risk metric we consider is the percentile metric. $\psi_\alpha = VaR_{0.5}$. $VaR_\alpha(Z) = \min_Z\{z|F_Z(z) \geq \alpha\}$. $\arg\max\{VaR_{0.5}[Z_1(\tau_1, u_1)]\} = a$

$$VaR_{0.5}[Z_{jt}(\boldsymbol{\tau}, \boldsymbol{u})] = \begin{cases} 1.25 & \boldsymbol{u} = (a, a) \\ 1 & \boldsymbol{u} = (a, b) \\ 1 & \boldsymbol{u} = (b, a) \\ 100 & \boldsymbol{u} = (b, b) \end{cases} \tag{B.10}$$

Assume, to the contrary, the VDN algorithm satisfy the RIGM theorem for the VaR metric. As $\arg\max_u\{VaR_{0.5}[Z_1(\tau_1, u_1)]\} = a$ and $\arg\max_u\{VaR_{0.5}[Z_2(\tau_2, u_2)]\} = a$, for the risk metric $\psi_{0.5} = VaR_{0.5}$, to satisfy the RIGM theorem, the optimal action that maximize the risk metric should be $(a, a)$. However, as it is shown in (B.10), the action maximizing $VaR_{0.5}Z(\tau, \boldsymbol{u})$ is $(b, b)$, rather than $(a, a)$, a contradiction. *We have shown that VDN does not satisfy the RIGM principle for the VaR metric.*

In this following, we show that QMIX does not satisfies the RIGM theorem for the $CVaR$ metric. QMIX is a method satisfying the IGM theorem. It learns $Q_{jt}(\tau, u) = \Phi(Q_1, ..., Q_n)$ to approximate the optimal policy of the state-action value function $Q(\tau, u)$, where $\Phi$ is a monotonic increasing

function with respect to $Q_i$. $\Phi = Q_1^3(\tau_1, u_1) + Q_2^3(\tau_2, u_2)$. Let's consider a simple one-step matrix game with two agents each with actions a and b. Using $\Phi$ as the factorization function, $Z_{jt}(\tau, u) = Z_1^3(\tau_1, u_1) + Z_2^3(\tau_2, u_2)$, where $Z_1 = Z_2$. $Z_{jt}$ could lead to incorrect estimation of the optimal actions for the risk metric $\psi_1 = CVaR_1$. $Z_i(\tau_i, a)$ and $Z_i(\tau_i, b)$ is defined as follows.

$$Z_i(\tau_i, a) = 2 \tag{B.11}$$

$$Z_i(\tau_i, b) = \begin{cases} 3 & 50\% \text{ of the time} \\ 0 & 50\% \text{ of the time} \end{cases} \tag{B.12}$$

Let's assume that, for action $a$, $Z_1(s, a) = 2$ for 100% of the time; for action $b$, $Z_1(s, b) = 3$ for 50% of the time and $Z_1(s, b) = 0$ for 50% of the time. Clearly $\psi_1[Z_1(\tau_1, a)] = CVaR_1[Z_1(\tau_1, a)]$ = 2 and $\psi_1[Z_1(\tau_1, b)] = CVaR_1[Z_1(\tau_1, b)] = 1.5$. If $Z_{jt}$ and $Z_i$ satisfy the RIGM theorem, then the optimal action for $Z_{jt}$ should be $(a, a) = (\arg\max_u \psi_1[Z_1(\tau_1, u_1)], \arg\max_u \psi_1[Z_2(\tau_2, u_2)])$. However, $\psi_1[Z_1^3(\tau, a)] = 8$ and $\psi_1[Z_1^3(\tau, b)] = 13.5$. $\arg\max_u \psi_1[Z_{jt}(\tau, u)] = (b, b)$ rather than $(a, a)$. A contradiction is found. *We have shown that methods satisfy the IGM principle do not guarantee the RIGM principle with risk metric $\psi_\alpha$.* $\square$

We show that DIGM factorization methods is insufficient to guarantee the satisfaction of the RIGM theorem.

**Theorem 2.** *Given a stochastic joint action-value function $Z_{jt}$, and a distributional factorization function $\Phi$ for the stochastic utilities which satisfy the DIGM theorem: the following distributional factorization:*

$$Z_{jt}(\tau, u) = \Phi(Z_1(\tau_1, u_1), ..., Z_n(\tau_n, u_n)) \tag{B.13}$$

*is insufficient to guarantee that $[Z_i]_{i=1}^n$ satisfy RIGM for $Z_{jt}(\tau, u)$ with risk metric $\psi_\alpha$.*

*Proof.* As far as we know, the DFAC framework and the ResZ method satisfies the DIGM principle. We first show that the DFAC framework does not guarantee the RIGM principle for the VaR metric and then show that ResZ does not guarantee the RIGM principle as well.

The DFAC framework learns factorize a joint return distribution $Z_{jt}$ using mean-shape decomposition which is defined as follows.

$$Z_{jt}(\tau, u) = \mathbb{E}[Z_{jt}(\tau, u)] + (Z_{jt}(\tau, u) - \mathbb{E}[Z_{jt}(\tau, u)]) \tag{B.14}$$
$$= Z_{mean}(\tau, u) + Z_{shape}(\tau, u) \tag{B.15}$$
$$= f(Q_1(\tau_1, u_1), ..., Q_n(\tau_n, u_n)) + h(Z_1(\tau_1, u_1), ..., Z_n(\tau_n, u_n)) \tag{B.16}$$
$$\tag{B.17}$$

where $f$ is factorization function for the deterministic utility. $f$ satisfy the IGM principle. The function $h$ models the shape of $Z_{jt}$. $h(Z_1(\tau_1, u_1), ..., Z_n(\tau_n, u_n)) = \sum_{i=1}^n (Z_i(\tau_i, u_i) - Q_i(\tau_i, u_i))$.

We consider a degenerated case where there is only one agent with two actions. The stochastic utility is defined as follows.

$$Z_1(\tau_1, a) = 5 \tag{B.18}$$

$$Z_1(\tau_1, b) = \begin{cases} 6 & 50\% \text{ of the time} \\ 0 & 50\% \text{ of the time} \end{cases} \tag{B.19}$$

$Q_1(\tau_1, a) = 5$ and $Q_1(\tau_1, b) = 3$. Let $f = 5^{Q_1(\tau_1, u_1)}$. In this case, for the $VaR_1$ risk measure $\psi_1 = VaR_1$, $\arg\max \psi_1[Z_{jt}(\tau, u)] = \arg\max[f(Q_1)] = a$. That is, the action $a$ maximize $\psi_1[Z_{jt}(\tau, u)]$. However, $\arg\max \psi_1[Z_1(\tau_1, u_1)] = b$ which is different from $a$. To satisfy the RIGM principle, these two actions should be equal. Thus, we have shown that *the DFAC framework does not guarantee the RIGM principle for the $VaR$ metric.*

We prove that ResZ does not guarantee the RIGM principle for the VaR metric through an example. The ResZ [8] algorithm is distributional factorization methods that satisfy the DIGM theorem but it cannot guarantee the RIGM principle.

ResZ [8] learns $Z_{jt}(\boldsymbol{\tau}, \boldsymbol{u}) = Z_{tot}(\boldsymbol{\tau}, \boldsymbol{u}) + w_r(\boldsymbol{\tau}, \boldsymbol{u}) Z_r(\boldsymbol{\tau}, \boldsymbol{u})$, where $Z_r(\boldsymbol{\tau}, \boldsymbol{u}) \leq 0$, $Z_{tot}(\boldsymbol{\tau}, \boldsymbol{u}) = \sum_{i=1}^{N} k_i Z_i(\tau_i, u_i)$ $k_i \geq 0$

We prove this theorem by providing an example. Let's assume a special case where there are two agents and $Z_r(\boldsymbol{\tau}, \boldsymbol{u}) = 0$. Then, $Z_{jt}(\boldsymbol{\tau}, \boldsymbol{u}) = Z_{tot}(\boldsymbol{\tau}, \boldsymbol{u}) = 2Z_1(\tau_1, u_1) + 2Z_2(\tau_2, u_2)$.

$$p(Z_i(\tau_i, a)) = \begin{cases} 50\% & Z_i = 0.25 \\ 50\% & Z_i = 1 \end{cases} \tag{B.20}$$

$$p(Z_i(\tau_i, b)) = \begin{cases} 50\% & Z_i = 0 \\ 50\% & Z_i = 100 \end{cases} \tag{B.21}$$

$$VaR_{0.5}[Z_{jt}(\boldsymbol{\tau}, \boldsymbol{u})] = \begin{cases} 2.5 & \boldsymbol{u} = (a, a) \\ 2 & \boldsymbol{u} = (a, b) \\ 2 & \boldsymbol{u} = (b, a) \\ 200 & \boldsymbol{u} = (b, b) \end{cases} \tag{B.22}$$

The risk metric we consider is $\psi_{0.5} = VaR_{0.5}$. $\arg\max\{VaR_{0.5}[Z_1(\tau_1, u_1)]\} = a$ and $\arg\max\{VaR_{0.5}[Z_2(\tau_2, u_2)]\} = a$, for ResZ, to satisfy the RIGM theorem, the optimal action that maximize the risk metric should be $(a, a)$. However, as it is shown in (B.22), the action maximizing $VaR_{0.5} Z_{jt}(\tau, \boldsymbol{u})$ is $(b, b)$, rather than $(a, a)$, a contradiction. $\qquad\square$

**Theorem 3.** *DRIMA [14], a risk-sensitive MARL algorithm, does not guarantee the RIGM principle.*

*Proof.* DRIMA [14] learns transformed action-value estimator $Z_{tran}(\boldsymbol{\tau}, \boldsymbol{u}, \alpha)$ to approximate the true value distribution function $Z_{jt}$. The quantile value $\theta_{tran}(\boldsymbol{\tau}, \boldsymbol{u}, \alpha)$ for $Z_{tran}(\boldsymbol{\tau}, \boldsymbol{u}, \alpha)$ is defined as $\theta_{tran}(\boldsymbol{\tau}, \boldsymbol{u}, \alpha) = Q_{mix}(\theta_1(\tau_1, u_1, \alpha), ..., \theta_n(\tau_n, u_n, \alpha))$, where $\theta_i(\tau_i, u_i, \alpha)$ is the $\alpha-$percentile for the stochastic utility $Z_i(\tau_i, u_i)$ of agent $i$.

We provide an example for DRIMA that does not satisfy the RIGM theorem for the risk-neutral case. For the risk-neutral case, DRIMA uniformly sample the quantile sample $\omega_j \in [0, 1]$, and the average value $\sum_{j=1}^{K} \theta(\boldsymbol{\tau}, \boldsymbol{u}, \omega_j)$ of the quantiles is used for action selection, where $K$ is the number of samples. When $K$ becomes infinite, DRIMA can be viewed as using the risk measure $CVaR_1$ (the expectation operation $\mathbb{E}$) to select actions. We will show that DRIMA does not satify the RIGM principle for the $CVaR_1$ metric.

Assuming there are two agents, each with two actions. $\theta_{tran}(\boldsymbol{\tau}, \boldsymbol{u}, \alpha) = \theta_1^3(\tau_1, u_1, \alpha) + \theta_2^3(\tau_2, u_2, \alpha)$. $Z_1 = Z_2$. $Z_{tran}$ could estimate sub-optimal actions for the risk metric $\psi_1 = CVaR_1$. The stochastic utilities $Z_1(\tau_1, a)$ and $Z_1(\tau_1, b)$ is defined as follows.

$$Z_1(\tau_1, a) = Z_2(\tau_2, a) = 3 \tag{B.23}$$

$$Z_1(\tau_1, b) = Z_2(\tau_2, b) = \begin{cases} 5 & 50\% \text{ of the time} \\ 0 & 50\% \text{ of the time} \end{cases} \tag{B.24}$$

Clearly $\psi_1[Z_1(s, a)] = 3$ and $\psi_1[Z_1(s, b)] = 2.5$. If DRIMA satisfies the RIGM theorem, then the risk-sensitive optimal action for $Z_{tran}$ should be $(a, a) = (\arg\max \psi_1[Z_1(\tau_1, u_1)], \arg\max \psi_1[Z_2(\tau_2, u_2)])$.

$$\psi_1[Z_{tran}(\boldsymbol{\tau}, \boldsymbol{u})] = \begin{cases} 54 & \boldsymbol{u} = (a, a) \\ 89.5 & \boldsymbol{u} = (a, b) \text{ or } (b, a) \\ 125 & \boldsymbol{u} = (b, b) \end{cases} \tag{B.25}$$

Clearly, $\arg\max \psi_1[Z_{tran}(\boldsymbol{\tau}, \boldsymbol{u})] = (b, b)$ rather than $(a, a)$. Thus, we have shown that DRIMA does not guarantee the RIGM theorem for the $CVaR_1$ metric. $\qquad\square$

## B.2 RiskQ

RiskQ satisfies the RIGM principle for the VaR and distorted expectation risk metrics (such as Wang, and CPW).

**Theorem 4.** *A stochastic joint state-action return distribution*

$$Z_{jt}(\boldsymbol{\tau}, \boldsymbol{u}) = \sum_{j=1}^{J} p_j(\boldsymbol{\tau}, \boldsymbol{u}, \omega_j) \delta_{\theta(\boldsymbol{\tau}, \boldsymbol{u}, \omega_j)} \tag{B.26}$$

$$\theta(\boldsymbol{\tau}, \boldsymbol{u}, \omega_j) = \sum_{i=1}^{n} k_i \theta_i(\tau_i, u_i, \omega_j) \tag{B.27}$$

*is distributional factorized by $[Z_i(\tau_i, u_i)]_{i=1}^{N}$ with risk metric $\psi_\alpha$, where $J$ is the number of Dirac Delta functions, $\delta_{\theta(\boldsymbol{\tau}, \boldsymbol{u}, \omega_j)}$ is a Dirac Delta function at $\theta(\boldsymbol{\tau}, \boldsymbol{u}, \omega_j) \in \mathbb{R}$, $\theta(\boldsymbol{\tau}, \boldsymbol{u}, \omega_j)$ is a quantile function with sample $\omega_j$, $p_j(\boldsymbol{\tau}, \boldsymbol{u}, \omega_j)$ is the corresponding probability for $\delta_{\theta(\boldsymbol{\tau}, \boldsymbol{u}, \omega_j)}$ of the estimated return, $\theta_i(\tau_i, u_i, \omega)$ is the $\omega - quantile$ function for $Z_i(\tau_i, u_i)$ and $k_i \geq 0$.*

*Proof.* Theorem show that if the above condition are satisfied, then $[Z_i(\tau_i, u_i)]_{i=1}^{N}$ satisfy the RIGM principle for $Z_{jt}(\boldsymbol{\tau}, \boldsymbol{u})$ with the risk metric $\psi_\alpha$. We will show that $\arg\max_u \psi_\alpha[Z_{jt}(\boldsymbol{\tau}, \boldsymbol{u})] = \bar{u}$, $\bar{u} = [\bar{u}_i]_{i=1}^{N}$, $\bar{u}_i = \arg\max_{u_i} \psi_\alpha[Z_i(\tau_i, u_i)]$.

We prove this theorem for the VaR and the Distorted Expectation risk metric (e.g., CVaR, CPW).

The VaR risk metric uses the $\alpha-$quantile to measure risk. $\psi_\alpha[Z_{jt}(\boldsymbol{\tau}, \boldsymbol{u})] = \theta(\boldsymbol{\tau}, \boldsymbol{u}, \alpha)$ and $\psi_\alpha[Z_i(\tau_i, u_i)] = \theta_i(\tau_i, u_i, \alpha)$

$$\psi_\alpha[Z_{jt}(\boldsymbol{\tau}, \bar{\boldsymbol{u}})] = \sum_{i=1}^{n} k_i \theta_i(\tau_i, \bar{u}_i, \alpha) \quad \text{(quantile definition)} \tag{B.28}$$

$$= \sum_{i=1}^{n} k_i \psi_\alpha[Z_i(\tau_i, \bar{u}_i)] \tag{B.29}$$

$$\geq \sum_{i=1}^{n} k_i \psi_\alpha[Z_i(\tau_i, u_i)] \quad (\bar{u}_i = \arg\max \psi_\alpha[Z_i(\tau_i, u_i)]) \tag{B.30}$$

$$= \sum_{i=1}^{n} k_i \theta_i(\tau_i, u_i, \alpha) \tag{B.31}$$

$$= \psi_\alpha[Z_{jt}(\boldsymbol{\tau}, \boldsymbol{u})] \tag{B.32}$$

Distorted expectation metrics, such as CVaR, CPW, and Wang, are weighted expectation of return distribution under a distortion function [9, 10, 24]. The distorted expectation of a random variable $Z$ under $g$ is defined as $\psi(Z) = \int_0^1 g'(w)\theta(w)dw$, where $g(w) \in [0, 1]$, $g'(w)$ is the derivative of $g(w)$. In the following, we show that if the above conditions are satisfied, then $[Z_i(\tau_i, u_i)]_{i=1}^{N}$ satisfy the RIGM principle for $Z_{jt}(\boldsymbol{\tau}, \boldsymbol{u})$ with the risk metric that is a distorted expectation metric.

$$\psi_\alpha[Z_{jt}(\boldsymbol{\tau}, \bar{\boldsymbol{u}})] = \int_0^1 g'(w)\theta(\boldsymbol{\tau}, \bar{\boldsymbol{u}}, w)dw \tag{B.33}$$

$$= \int_0^1 g'(w)\sum_{i=1}^n k_i\theta_i(\tau_i, \bar{u}_i, w)dw \tag{B.34}$$

$$= \sum_{i=1}^n k_i\int_0^1 g'(w)\theta_i(\tau_i, \bar{u}_i, w)dw \tag{B.35}$$

$$= \sum_{i=1}^n k_i\psi_\alpha[Z_i(\tau_i, \bar{u}_i)] \tag{B.36}$$

$$\geq \sum_{i=1}^n k_i\psi_\alpha[Z_i(\tau_i, u_i)] \quad (\bar{u}_i = \arg\max\psi_\alpha[Z_i(\tau_i, u_i)]) \tag{B.37}$$

$$= \sum_{i=1}^n k_i\int_0^1 g'(w)\theta_i(\tau_i, u_i, w)dw \tag{B.38}$$

$$= \int_0^1 g'(w)\sum_{i=1}^n k_i\theta_i(\tau_i, u_i, w)dw \tag{B.39}$$

$$= \int_0^1 g'(w)\theta(\boldsymbol{\tau}, \boldsymbol{u}, w)dw \tag{B.40}$$

$$= \psi_\alpha[Z_{jt}(\boldsymbol{\tau}, \boldsymbol{u})] \tag{B.41}$$

$\square$

We have shown that by modeling the quantiles of the stochastic $Z_{jt}$ through weighted sum of quantiles of $[Z_i]_{i=1}^n$. $[Z_i]_{i=1}^n$ satisfied the RIGM theorem with risk metrics such as VaR, CVaR, Wang, and CPW, etc.

## B.3  RiskQ-QMIX

RiskQ suffers from representation limitations that it can model the weighted sum relationships among quantiles. RiskQ-QMIX replaces the mixer of RiskQ from a simple attention module to QMIX. It can model the monotonic relationship among quantiles. We show that it satisfies the RIGM principle for the VaR metric.

**Theorem 5.** *A risk-aware stochastic joint state-action return*

$$Z_{jt}(\boldsymbol{\tau}, \boldsymbol{u}) = \sum_{j=1}^J p_j(\boldsymbol{\tau}, \boldsymbol{u})\delta_{\theta(\boldsymbol{\tau}, \boldsymbol{u}, w_j)} \tag{B.42}$$

$$\theta(\boldsymbol{\tau}, \boldsymbol{u}, w_j) = Q_{mix}(\theta_1(\tau_1, u_1, w_j), ...\theta_n(\tau_n, u_n, w_j)) \tag{B.43}$$

*is distributional factorized by $[Z_i(\tau_i, u_i)]_{i=1}^N$ with risk metric $VaR_\alpha$, where $J$ is the number of Dirac Delta functions, $\delta_{\theta(\boldsymbol{\tau}, \boldsymbol{u}, w_j)}$ is a Dirac Delta function at $\theta(\boldsymbol{\tau}, \boldsymbol{u}, w_j) \in \mathbb{R}$, $\theta(\boldsymbol{\tau}, \boldsymbol{u}, w_j)$ is a quantile function with quantile sample $w_j$, $p_j(\boldsymbol{\tau}, \boldsymbol{u})$ is the corresponding probability for $\delta_{\theta(\boldsymbol{\tau}, \boldsymbol{u}, w_j)}$ of the estimated return, $\theta_i(\tau_i, u_i, w)$ is the $w-$quantile for $Z_i(\tau_i, u_i)$.*

*Proof.* Theorem show that if the above condition are satisfied, then $[Z_i(\tau_i, u_i)]_{i=1}^N$ satisfy the RIGM principle for $Z_{jt}(\boldsymbol{\tau}, \boldsymbol{u})$ with the risk metric $VaR_\alpha$. We will show that $\arg\max_u \psi_\alpha[Z_{jt}(\boldsymbol{\tau}, \boldsymbol{u})] = \bar{u}$, $\bar{u} = [\bar{u}_i]_{i=1}^N$, $\bar{u}_i = \arg\max_{u_i} \psi_\alpha[Z_i(\tau_i, u_i)]$.

The VaR risk metric uses $\alpha-$quantile of a random variable to measure risk of the variable. $VaR_\alpha[Z_{jt}(\boldsymbol{\tau}, \boldsymbol{u})] = \theta(\boldsymbol{\tau}, \boldsymbol{u}, \alpha)$ and $VaR_\alpha[Z_i(\tau_i, u_i)] = \theta_i(\tau_i, u_i, \alpha)$

$$VaR_\alpha[Z_{jt}(\boldsymbol{\tau}, \bar{\boldsymbol{u}})] = Q_{mix}(\theta_1(\tau_1, \bar{u}_1, \alpha), ...\theta_n(\tau_n, \bar{u}_n, \alpha)) \quad \text{(quantile definition)} \tag{B.44}$$

$$\geq Q_{mix}(\theta_1(\tau_1, u_1, \alpha), ...\theta_n(\tau_n, u_n, \alpha)) \quad (\bar{u}_i = \arg\max VaR_\alpha[Z_i(\tau_i, u_i)]) \tag{B.45}$$

$$= \psi_\alpha[Z_{jt}(\boldsymbol{\tau}, \boldsymbol{u})] \tag{B.46}$$

(B.45) is satisfied due to the following reasons. $\bar{u}_i = \arg\max VaR_\alpha[Z_i(\tau_i, u_i)]$, thus $\theta_i(\tau_i, \bar{u}_i, \alpha) \geq \theta_i(\tau_i, u_i, \alpha)$. As $Q_{mix}(\theta_1(\tau_1, u_1, \alpha), ...\theta_n(\tau_n, u_n, \alpha))$ is monotonic increase with respect to $\theta_i(\tau_i, u_i, \alpha)$, $Q_{mix}(\theta_1(\tau_1, \bar{u}_1, \alpha), ...\theta_n(\tau_n, \bar{u}_n, \alpha)) \geq Q_{mix}(\theta_1(\tau_1, u_1, \alpha), ...\theta_n(\tau_n, u_n, \alpha))$

□

## B.4 RiskQ-Residual

RiskQ-QMIX suffers from representation limitation that it can model the monotonic relationship among quantiles only. It cannot model non-monotonic relationship among quantiles. ResZ [8] decomposes a stochastic joint return distribution $Z_{jt}(\boldsymbol{\tau}, \boldsymbol{u})$ into a main function $Z_{tot}(\boldsymbol{\tau}, \boldsymbol{u})$ and a residual function $Z_r(\boldsymbol{\tau}, \boldsymbol{u})$. The main function $Z_{tot}(\boldsymbol{\tau}\boldsymbol{u})$ share the same optimal policy as $Z_{jt}(\boldsymbol{\tau}, \boldsymbol{u})$. It is show that ResZ satisfies the DIGM principle without representation limitations.

Inspired by ResZ, we decompose a quantile function into its main quantile function $\sum_{i=1}^{n} k_i \theta_i(\tau_i, u_i, w_j)$ and residual quantile function $\theta_r(\boldsymbol{\tau}, \boldsymbol{u}, w_j)$ with a mask function $m_\alpha(\boldsymbol{\tau}, \boldsymbol{u})$. We show that RiskQ-Residual satisfies the RIGM principle for the VaR and distorted expectations metrics without representation limitations.

**Theorem 6.** *A stochastic joint state-action return*

$$Z_{jt}(\boldsymbol{\tau}, \boldsymbol{u}) = \sum_{j=1}^{J} p_j(\boldsymbol{\tau}, \boldsymbol{u}) \delta_{\theta(\boldsymbol{\tau}, \boldsymbol{u}, w_j)} \tag{B.47}$$

$$\theta(\boldsymbol{\tau}, \boldsymbol{u}, w_j) = \sum_{i=1}^{n} k_i \theta_i(\tau_i, u_i, w_j) + m_\alpha(\boldsymbol{\tau}, \boldsymbol{u}) \theta_r(\boldsymbol{\tau}, \boldsymbol{u}, w_j) \tag{B.48}$$

*is distributional factorized by $[Z_i(\tau_i, u_i)]_{i=1}^{N}$ with risk metric $\psi_\alpha$, where $\theta_r(\boldsymbol{\tau}, \boldsymbol{u}, w) \leq 0$, the mask function $m_\alpha(\boldsymbol{\tau}, \boldsymbol{u}) = 0$ when $\boldsymbol{u} = \bar{\boldsymbol{u}}$, otherwise 1, J is the number of Dirac Delta functions, $\delta_{\theta(\boldsymbol{\tau}, \boldsymbol{u}, w_j)}$ is a Dirac Delta function at percentile $\theta(\boldsymbol{\tau}, \boldsymbol{u}, w_j) \in \mathbb{R}$, $\theta(\boldsymbol{\tau}, \boldsymbol{u}, w_j)$ is a quantile function with quantile $w_j$, $p_j(\boldsymbol{\tau}, \boldsymbol{u})$ is the corresponding probability for $\delta_{\theta(\boldsymbol{\tau}, \boldsymbol{u}, w_j)}$ of the estimated return, $\theta_i(\tau_i, u_i, w)$ is the $w-$quantile for the stochastic utility function $Z_i(\tau_i, u_i)$ and $k_i \geq 0$.*

*Proof.* Theorem 6 shows that if the above conditions are satisfied, then $[Z_i(\tau_i, u_i)]_{i=1}^{N}$ satisfy the RIGM principle for $Z_{jt}(\boldsymbol{\tau}, \boldsymbol{u})$ with the risk metric $\psi_\alpha$. We will show that $\arg\max_{\boldsymbol{u}} \psi_\alpha[Z_{jt}(\boldsymbol{\tau}, \boldsymbol{u})] = \bar{\boldsymbol{u}}, \bar{\boldsymbol{u}} = [\bar{u}_i]_{i=1}^{N}, \bar{u}_i = \arg\max_{u_i} \psi_\alpha[Z_i(\tau_i, u_i)]$.

For the VaR metric

$$\psi_\alpha[Z_{jt}(\boldsymbol{\tau}, \bar{\boldsymbol{u}})] = \sum_{i=1}^{n} k_i \theta_i(\tau_i, \bar{u}_i, \alpha) + m_\alpha(\boldsymbol{\tau}, \bar{\boldsymbol{u}})\theta_r(\boldsymbol{\tau}, \bar{\boldsymbol{u}}, \alpha) \tag{B.49}$$

$$= \sum_{i=1}^{n} k_i \theta_i(\tau_i, \bar{u}_i, \alpha) \quad (m_r(\boldsymbol{\tau}, \bar{\boldsymbol{u}}, \alpha) = 0) \tag{B.50}$$

$$= \sum_{i=1}^{n} k_i \psi_\alpha[Z_i(\tau_i, \bar{u}_i)] \quad \text{VaR definition} \tag{B.51}$$

$$\geq \sum_{i=1}^{n} k_i \psi_\alpha[Z_i(\tau_i, u_i)] \quad (\bar{u}_i = \arg\max \psi_\alpha[Z_i(\tau_i, u_i)]) \tag{B.52}$$

$$= \sum_{i=1}^{n} k_i \theta_i(\tau_i, u_i, \alpha) \tag{B.53}$$

$$\geq \sum_{i=1}^{n} k_i \theta_i(\tau_i, u_i, \alpha) + m_r(\boldsymbol{\tau}, \boldsymbol{u}, \alpha)\theta_r(\boldsymbol{\tau}, \boldsymbol{u}, \alpha) \tag{B.54}$$

$$= \psi_\alpha[Z_{jt}(\boldsymbol{\tau}, \boldsymbol{u})] \tag{B.55}$$

(B.54) because $\theta_r(\boldsymbol{\tau}, \boldsymbol{u}, w_j) \leq 0$ and $m_r(\boldsymbol{\tau}, \boldsymbol{u}, \alpha) = 1$

(B.49) to (B.55) mean that $\bar{u} = [\bar{u}_i]_{i=1}^N$ maximizes $\psi_\alpha[Z_{jt}(\alpha, \boldsymbol{\tau}, \boldsymbol{u})]$ for the VaR metric. Thus $[Z_i(\tau_i, u_i)]_{i=1}^N$ satisfies RIGM for $Z_{jt}(\boldsymbol{\tau}, \boldsymbol{u})$ with for the VaR risk metric $\psi_\alpha$.

For distorted expectation metrics $\psi_\alpha$ such as Wang, CVaR, and CPW, and show that $[Z_i(\tau_i, u_i)]_{i=1}^N$ satisfy the RIGM theorem for $\psi_\alpha$ as follow.

$$\psi_\alpha[Z_{jt}(\boldsymbol{\tau}, \bar{\boldsymbol{u}})] = \int_0^1 g'(w)\theta(\boldsymbol{\tau}, \bar{\boldsymbol{u}}, w)dw \tag{B.56}$$

$$= \int_0^1 g'(w)[\sum_{i=1}^{n} k_i \theta_i(\tau_i, \bar{u}_i, w) + m_r(\boldsymbol{\tau}, \bar{\boldsymbol{u}}, w)\theta_r(\boldsymbol{\tau}, \bar{\boldsymbol{u}}, w)]dw \tag{B.57}$$

$$= \int_0^1 g'(w) \sum_{i=1}^{n} k_i \theta_i(\tau_i, \bar{u}_i, w)dw \quad (m_r(\boldsymbol{\tau}, \bar{\boldsymbol{u}}, w) = 0) \tag{B.58}$$

$$= \sum_{i=1}^{n} k_i \int_0^1 g'(w)\theta_i(\tau_i, \bar{u}_i, w)dw \tag{B.59}$$

$$= \sum_{i=1}^{n} k_i \psi_\alpha[Z_i(\tau_i, \bar{u}_i)] \quad (\bar{u}_i = \arg\max \psi_\alpha[Z_i(\tau_i, u_i)]) \tag{B.60}$$

$$\geq \sum_{i=1}^{n} k_i \psi_\alpha[Z_i(\tau_i, u_i)] \tag{B.61}$$

$$= \sum_{i=1}^{n} k_i \int_0^1 g'(w)\theta_i(\tau_i, u_i, w)dw \tag{B.62}$$

$$= \int_0^1 g'(w) \sum_{i=1}^{n} k_i \theta_i(\tau_i, u_i, w)dw \tag{B.63}$$

$$\geq \int_0^1 g'(w)[\sum_{i=1}^{n} k_i \theta_i(\tau_i, u_i, w) + m_r(\boldsymbol{\tau}, \boldsymbol{u}, w)\theta_r(\boldsymbol{\tau}, \boldsymbol{u}, w)]dw \tag{B.64}$$

$$= \int_0^1 g'(w)\theta(\boldsymbol{\tau}, \boldsymbol{u}, w)dw \tag{B.65}$$

$$= \psi_\alpha[Z_{jt}(\boldsymbol{\tau}, \boldsymbol{u})] \tag{B.66}$$

(B.56) to (B.65) mean that $\bar{u} = [\bar{u}_i]_{i=1}^N$ maximizes $\psi_\alpha[Z_{jt}(\alpha, \boldsymbol{\tau}, \boldsymbol{u})]$ for distorted metrics such as $CVaR_\alpha$. Thus $[Z_i(\alpha, \tau_i, u_i)]_{i=1}^N$ satisfies RIGM for $Z_{jt}(\alpha, \boldsymbol{\tau}, \boldsymbol{u})$ with for distorted expectation metrics.

$\square$

## B.5 RiskQ-Residual-QMIX

RiskQ-Residual-QMIX model the main quantile function as $Q_{mix}(\theta_1(\tau_1, \bar{u}_1, \alpha), ...\theta_n(\tau_n, \bar{u}_n, \alpha))$. We show that it satisfies the RIGM principle for the VaR metric.

**Theorem 7.** *A stochastic joint state-action return*

$$Z_{jt}(\boldsymbol{\tau}, \boldsymbol{u}) = \sum_{j=1}^J p_j(\boldsymbol{\tau}, \boldsymbol{u})\delta_{\theta(\boldsymbol{\tau}, \boldsymbol{u}, w_j)} \tag{B.67}$$

$$\theta(\boldsymbol{\tau}, \boldsymbol{u}, w_j) = Q_{mix}(\theta_1(\tau_1, \bar{u}_1, \alpha), ...\theta_n(\tau_n, \bar{u}_n, \alpha)) + m_\alpha(\boldsymbol{\tau}, \boldsymbol{u})\theta_r(\boldsymbol{\tau}, \boldsymbol{u}, w_j) \tag{B.68}$$

*is distributional factorized by $[Z_i(\tau_i, u_i)]_{i=1}^N$ with risk metric $\psi_\alpha$, where $\theta_r(\boldsymbol{\tau}, \boldsymbol{u}, w) \leq 0$, the mask function $m_\alpha(\boldsymbol{\tau}, \boldsymbol{u}) = 0$ when $\boldsymbol{u} = \bar{u}$, otherwise 1, J is the number of Dirac Delta functions, $\delta_{\theta(\boldsymbol{\tau}, \boldsymbol{u}, w_j)}$ is a Dirac Delta function at percentile $\theta(\boldsymbol{\tau}, \boldsymbol{u}, w_j) \in \mathbb{R}$, $\theta(\boldsymbol{\tau}, \boldsymbol{u}, w_j)$ is a quantile function with quantile $w_j$, $p_j(\boldsymbol{\tau}, \boldsymbol{u})$ is the corresponding probability for $\delta_{\theta(\boldsymbol{\tau}, \boldsymbol{u}, w_j)}$ of the estimated return, $\theta_i(\tau_i, u_i, w)$ is the w−quantile for the stochastic utility function $Z_i(\tau_i, u_i)$.*

For the VaR metric

$$\psi_\alpha[Z_{jt}(\boldsymbol{\tau}, \bar{\boldsymbol{u}})] = Q_{mix}(\theta_1(\tau_1, \bar{u}_1, \alpha), ...\theta_n(\tau_n, \bar{u}_n, \alpha)) + m_\alpha(\boldsymbol{\tau}, \bar{\boldsymbol{u}})\theta_r(\boldsymbol{\tau}, \bar{\boldsymbol{u}}, \alpha) \quad \text{VaR definition} \tag{B.69}$$

$$= Q_{mix}(\theta_1(\tau_1, \bar{u}_1, \alpha), ...\theta_n(\tau_n, \bar{u}_n, \alpha))) \quad (m_r(\boldsymbol{\tau}, \bar{\boldsymbol{u}}, \alpha) = 0) \tag{B.70}$$

$$\geq Q_{mix}(\theta_1(\tau_1, u_1, \alpha), ...\theta_n(\tau_n, u_n, \alpha))) \quad (\bar{u}_i = \arg\max \psi_\alpha[Z_i(\tau_i, u_i)]) \tag{B.71}$$

$$\geq Q_{mix}(\theta_1(\tau_1, u_1, \alpha), ...\theta_n(\tau_n, u_n, \alpha))) + m_r(\boldsymbol{\tau}, \boldsymbol{u}, \alpha)\theta_r(\boldsymbol{\tau}, \boldsymbol{u}, \alpha) \tag{B.72}$$

$$= \psi_\alpha[Z_{jt}(\boldsymbol{\tau}, \boldsymbol{u})] \quad \text{VaR definition} \tag{B.73}$$

(B.72) because $\theta_r(\boldsymbol{\tau}, \boldsymbol{u}, w_j) \leq 0$ and $m_r(\boldsymbol{\tau}, \boldsymbol{u}, \alpha) = 1$

## B.6 Algorithm and Neural Networks

### B.6.1 Neural Networks

We provide the detailed riskq network framework in Figure 1. It contains (a): the agent network, (b): the overall framework of RiskQ, and (c): the hybrid network diagram of RiskQ.

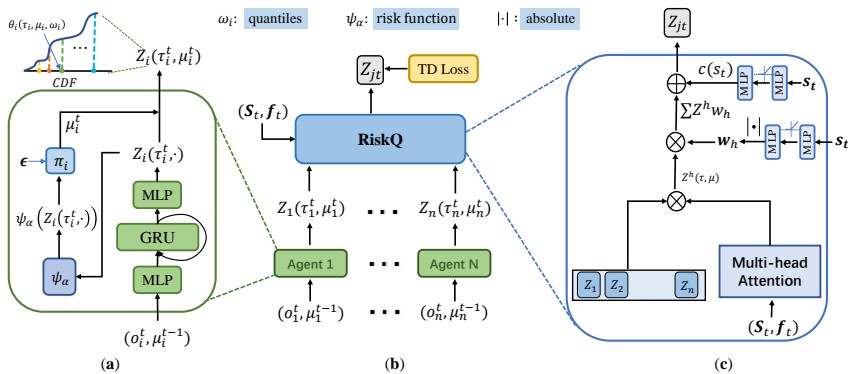

Figure 1: RiskQ framework

As shown in Figure 1, the agent network takes the observation-action sequence $(o_i^t, u_i^{t-1})$ from agent $i$ at time step $t$, and channels it through a multi-layer perception architecture, specifically structured

as MLP-GRU-MLP, to generate the distribution $Z_i$. Then, we can obtain the $\psi(Z_i)$ through the risk function, and the action $u_i^t$ is selected through the $\epsilon - greedy$ strategy, finally obtaining the distribution $Z_i(\tau_i^t, u_i^t)$. Each agent produces the distribution $Z_i(\tau_i^t, u_i^t)$ through the agent network, and then generates $z_{jt}$ through the RiskQ Mixer. The RiskQ Mixer combines the quantile sample (cumulative probability) $w$ of the distribution $Z_i$, with the multi-layer attention mechanism and the multi-layer perception network, and finally obtains the quantile value $\theta(\boldsymbol{\tau}, \boldsymbol{\mu}, \omega)$ of $Z_{jt}(\boldsymbol{\tau}, \boldsymbol{\mu})$, which is represented as the weighted sum of the quantile values of stochastic utility $\sum_n^i = k_i \theta_i(\tau, \mu_i, \omega)$.

### B.6.2 Algorithm

The RiskQ algorithm is described in Algorithm 1.

---

**Algorithm 1** The RiskQ Algorithm

---

**Require:** risk parameter: $\psi_\alpha$;
**Require:** Initialize parameters $\theta$ of the network of agent, risk operator and RiskQ neural networks;
 1: **for** $e \in \{1, \ldots, m$ episodes$\}$ **do**
 2:     Start a new episode;
 3:    **while** episode_is_not_end **do**
 4:       **for** agent $i$ **do**
 5:          Estimate the agent's utility $Z_i^t(\tau_i^t, u_i^t)$;
 6:          Calculate Risk-sensitive value $\psi_\alpha[Z(\tau_i^t, u_i^t)]$ ;
 7:          Get the action $\bar{u}_i^t = \arg\max_u \psi_\alpha[Z(\tau_i^t, u)]$;
 8:       **end for**
 9:       Execute $\bar{u}^t$, obtain global reward $r^t$ and the next state $s'$
10:       update replay buffer $\mathcal{D}$;
11:       **if** it is time to update **then**
12:          Sample a batch $\mathcal{D}'$ from replay buffer $\mathcal{D}$;
13:          For each sample in $\mathcal{D}'$, calculate $Z_{jt}$, $\psi_\alpha[Z_{jt}]$.
14:          Update $\theta$ by minimizing the Huber quantile loss;
15:       **end if**
16:    **end while**
17: **end for**

---

## C   Evaluation

We study the performance of RiskQ on risk-sensitive games (Multi-agent cliff and Car following games), the StarCraft II Multi-Agent Challenge benchmark (SMAC) [16]. RiskQ can obtain promising performance for risk-sensitive and risk-neutral scenarios. Ablation studies reveal the importance of adhering to the RIGM principle for achieving good performance. Additionally, we have examined the impact of functional representations, distribution utilities, risk metrics and risk levels.

In this secion, we seek to answer the following questions.

1. What are the benefits brought by RiskQ?
2. What will happen if we do not satisfy the RIGM principle?
3. What's the impact of representation limitations for RiskQ?
4. What are alternative designs for RiskQ?

We answer these questions through.

1. Evaluating RiskQ with risk-sensitive environments (Sec. C.2 C.3 C.4), and evaluating RiskQ with risk-neutral environment (Sec. C.4). We find that RiskQ can obtain promising results for all these environments. And we show that RiskQ allow user specify different risk-sensitive object to optimize a different goal (Figure 9).

2. We evaluate several cases where some agents act according to another risk-metric in Figure 14. We show that it is necessary to for agent act consistently according to the same risk-metric.

3. We evaluate different variants of RiskQ, which has better representation power in Sec C.6.2. We show the the representation limitations of RiskQ does not impact its performance significantly.

4. We evaluate different implementations of the mixer functions and utility functions.

## C.1 Experimental Setup

We select three categories of MARL value factorization methods for comparison: (i) Expected value factorization methods: QMIX [5], QTran [6], QPlex [7], CW QMIX [15], ResQ [8]; (ii) Risk-neutral stochastic value factorization methods: DMIX [22] and ResZ [8]; (iii) Risk-sensitive stochastic value factorization methods: RMIX [13] and DRIMA [14]. For robustness, each experiment is conducted at least 5 times with different random seeds. In general, the configuration of RiskQ follows the setup of Weighted QMIX and ResQ. By default, the risk metric used in RiskQ is $Wang_{0.75}$, indicating a risk-averse preference.

The work used for comparison is listed as follows.

Table 1: Baseline algorithms

| Algorithms | Brief Description |
|---|---|
| QMIX[2] [5] | Facilitates a monotonic combination of individual agent utilities. |
| QPlex[3] [7] | Learns a mixer of advantage functions and state value functions. |
| CW QMIX[4] [15] | A centrally-weighted version of QMIX |
| ResQ[5] / ResZ [8] | Converts the joint value function/distribution into main function plus residual function. |
| DMIX[6] [22] | Integrates distributional RL with QMIX |
| RMIX[7] [13] | Incorporates CVaR-optimized policies into QMIX, using present and past observations as supplementary inputs for each RMIX agent. |
| DRIMA[8] [14] | Separates cooperation risk from environmental risk, employing IQN as each agent's utility. |

We implement each algorithm based on their open-source repositories to carry out performance analyses, with hyperparameters consistent with those in PyMARL. RiskQ is also developed within the PyMARL framework, following the setup of WQMIX and ResQ. For DRIMA, the default configuration in standard scenarios is cooperative risk-seeking and environmental risk-neutral. For RiskQ, unless otherwise specified, the following default configuration is adopted: $Wang_{0.75}$ is used as the risk measurement. QR-DQN is used to model per-agent's stochastic utility, and the quantile number is set to 32. The RMSProp optimizer is employed with a learning rate of 0.001. Batch size and buffer size are set to 32 and 5000, respectively. RiskQ uses TD-lambda learning with $\lambda = 0.6$. The $\epsilon$ used in $\epsilon$-greedy annealed from 1 to 0.05 within 100K time steps. For robustness, each experiment is conducted at least 5 times with different random seeds. Experiments are carried out on a clusters consists of multiple NVIDIA GeForce RTX 3090 GPUs.

## C.2 Multi-Agent Cliff Navigation

As depicted in Figure 2, in Multi-Agent Cliff Navigation (MACN) which was introduced in [13], two agents must navigate in a grid-like world to reach the goal without falling into cliff. The agent receives a -1 reward at each time step. If any agent reaches the goal individually, a -0.5 reward is given to the agents. They will receive 00 reward if they reach the goal together. An episode is finished once the goal is reached by two agents together or any agent falls into cliff (reward -100). We depicted the test return of each learning algorithm in two MACN scenarios: $4 \times 11$ grid map and $4 \times 15$ grid map.

---

[2]https://github.com/oxwhirl/pymarl
[3]https://github.com/wjh720/QPLEX
[4]https://github.com/oxwhirl/wqmix
[5]https://github.com/xmu-rl-3dv/ResQ
[6]https://github.com/j3soon/dfac
[7]https://github.com/yetanotherpolicy/rmix
[8]https://github.com/osilab-kaist/

Here, we provide some additional baseline algorithms for performance comparison. In this environment, we employed the default configuration of RiskQ. For DRIMA, We considered two configurations in detail, namely DRIMA_sn and DRIMA_sa. DRIMA_sn represents a configuration that adopts a cooperative risk-seeking and environmental risk-neutral policy, while DRIMA_sa implies a configuration utilizing a cooperative risk-seeking and environmental risk-neutral policy. Cooperative risk-seeking prefer-

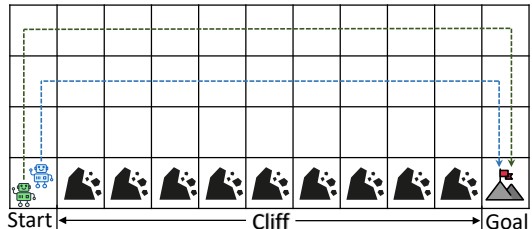

Figure 2: Multi-Agent Cliff Navigation

ence is adopted due to the collaborative nature of the MACN environment, as described in DRIMA, where cooperative risk-seeking can promote cooperation among agents. However, MACN is an environment with risks, thereby we hold preference for being neutral and averse towards environmental risk, respectively. As illustrated in Figure 3 (a) and (b), RiskQ achieves the optimal performance in both scenarios, outperforming risk-sensitive methods: RMIX, DRIMA_sa and DRIMA_sn. This indicates that learning policies which satisfy the RIGM principle could lead to promising results.

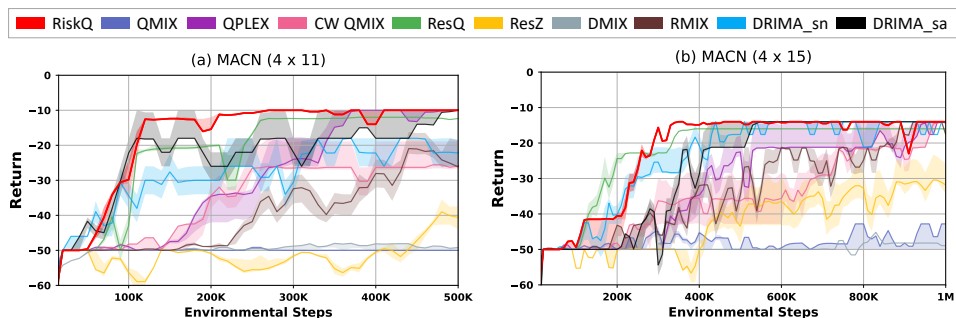

Figure 3: The return of two MACN scenarios with different map sizes.

## C.3 Multi-Agent Car Following Game

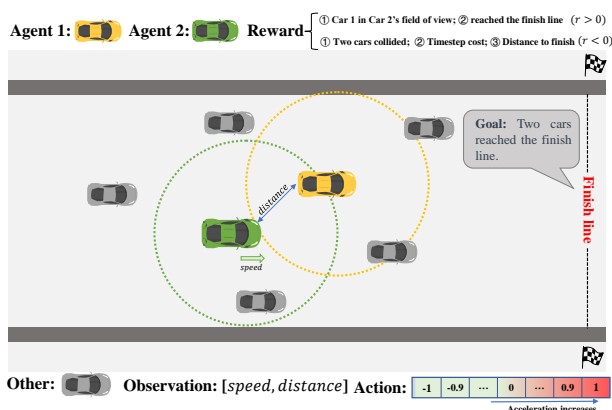

Figure 4: Multi-Agent Car Following Game

We design Multi-Agent Car Following (MACF) Game, which is adapted from the single agent risk-sensitive environment [35]. As depicted in Figure 4, in MACF, there are two agents, each controlling one car, with the task of one car following the other to reach a goal. Each car can observe the current position and speed of other cars within its observation range. The agent has a fixed action space which determines its acceleration. At each time step, agent will receive a negative reward. And the cars will *crash* with some probability if their speed exceed a speed-threshold, and a negative reward is given to the agents. To adapt the game to cooperative MARL, agents move within each other's observation will receive a positive reward. Once the agents reach the goal together, a big reward is given to them and the episode is terminated.

In MACF, rewards stem from the following sources:

1. Individual rewards when a single car reaches its destination.

2. Collaborative rewards when two cars arrive at their destination together.

3. Agents seeing each other within their observation range.

4. Negative reward at each time step.

5. Rewards based on the distance between cars to the destination.

6. Crash cost.

The configuration of these rewards impacts the environment's preference for different policies. For the MACF environment, we developed two different scenarios for evaluation: a pessimistic scenario and an optimistic scenario. The source code of the MACF environment can be obtained from the supplementary materials.

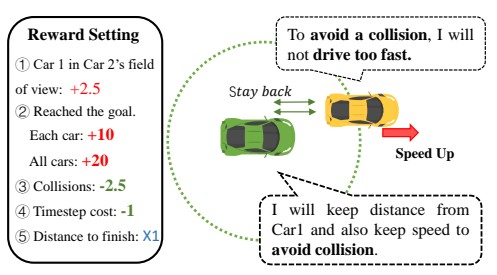

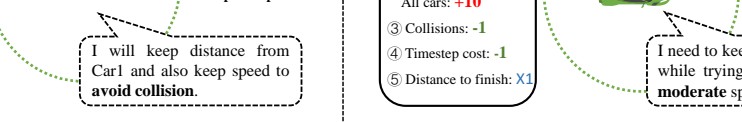


<div>

**Reward Setting**
① Car 1 in Car 2's field of view: **+2.5**
② Reached the goal.
  Each car: **+10**
  All cars: **+20**
③ Collisions: **-2.5**
④ Timestep cost: **-1**
⑤ Distance to finish: X1

To **avoid a collision**, I will not **drive too fast**.

*Stay back*

**Speed Up**

I will keep distance from Car1 and also keep speed to **avoid collision**.

</div>
<div>

**Reward Setting**
① Car 1 in Car 2's field of view: **+1**
② Reached the goal.
  Each car: **+5**
  All cars: **+10**
③ Collisions: **-1**
④ Timestep cost: **-1**
⑤ Distance to finish: X1

I'll try to keep a moderate pace to **balance speed and safety**.

*follow*

**Slow**

I need to keep up with Car1 while trying to maintain a **moderate** speed

</div>
</div>

Figure 5: Pessimistic scenario        Figure 6: Optimistic scenario

In the pessimistic scenario, the reward settings, as shown in Figure 5, are adjusted to have a crash probability of 1.0 after exceeding the speed limit. In such an environment, agents should learn policies that can reach the destination as quickly as possible, while ensuring safety. RiskQ in this scenario adopts $CVaR_{0.2}$ as the risk measurement, which represents a risk-averse preference. To ensure fairness, we adjusted the environmental risk of DRIMA to be averse. We have implemented risk-averse mode for DRIMA, then we conducted experiments of DRIMA with both risk-seeking and risk-neutral cooperative risk. From the results shown in Figure 7 (a) and (b), it is evident that RiskQ outperforms other approaches in terms of both return and crash.

In the optimistic scenario, the reward settings, as depicted in Figure 6, are defined such that the probability of a crash is $0.4 \times the\_amout\_of\_overspeed$. The settings in this environment requires cars to achieve a balance between speed and safety. RiskQ in this scenario adopts $CVaR_{1.0}$ as the risk measurement, which represents a risk-neutral risk preference. To ensure fairness, we adjusted the environmental risk of DRIMA to be neutral. We conducted experiments with both risk-seeking and risk-neutral cooperative risk. From the results shown in Figure 7 (c) and (d), it is evident that RiskQ outperforms other approaches in terms of both return and crash.

## C.4 StarCraft II

We examine the performance of RiskQ and several other algorithms using the StarCraft II Multi-Agent Challenge (SMAC), a widely recognized benchmark in the field of MARL. The SMAC environment consists of two competing teams of agents engaged in combat scenarios. One team is controlled by the carefully handcrafted built-in game artificial intelligence, the other team comprises agents guided by decentralized policies learned through MARL algorithms. Consistent with many prior works [8, 13], we set the AI level for controlling the enemy team in SMAC to 7 (very difficult). Each agent possesses a circular observation range and can engage in close-range combat with nearby enemies. At each time step, the agents make decisions to either move or perform actions related to attacking or healing. The rewards are affected by the damage inflicted on enemy units, with an additional reward given for eliminating all enemy units. The highest achievable reward is normalized to 20. We use the default reward schema provided by the SMAC benchmark to maintain consistency across experiments.

Following the evaluation protocol of DRIMA for risk-averse SMAC, we first study the performance of RiskQ in an explorative, a dilemmatic, and a random setting for the 3s_vs_5z, MMM2, and 2c_vs_64zg scenario. This setting is described as follows.

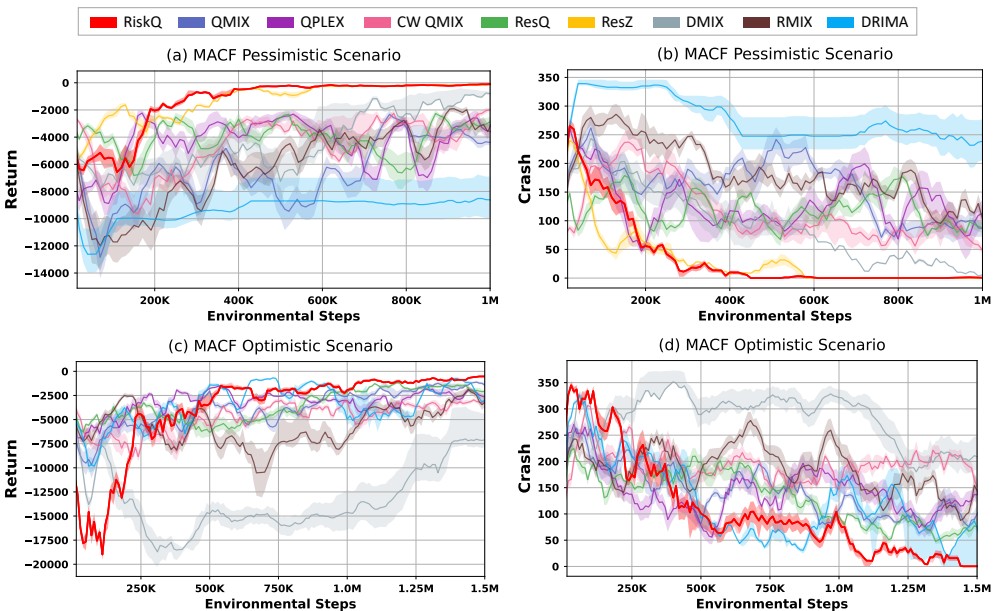

Figure 7: The return and the average number of crashes of two MACF scenarios. (a) Return of the pessimistic scenario. (b) The average number of crashes of the pessimistic scenario. (c) Return of the optimistic scenario. (d) The average number of crashes of the optimistic scenario.

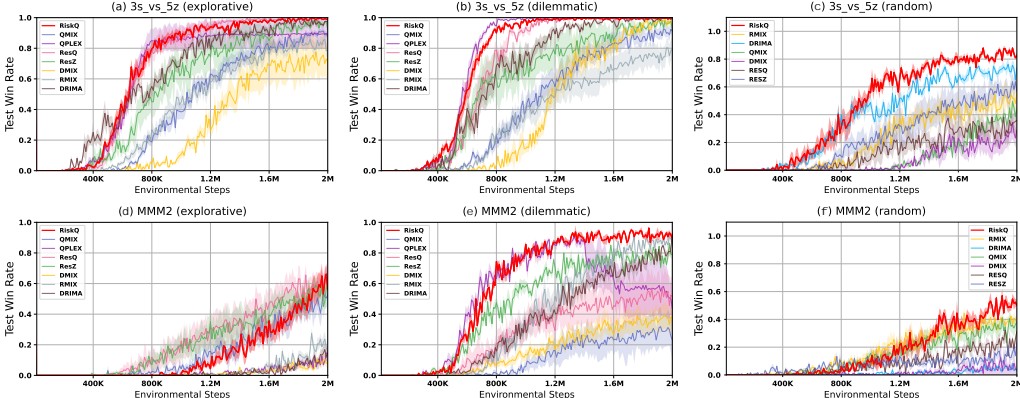

Figure 8: Performance Comparison in Explorative, Dilemmatic, and Random SMAC Scenarios.

1. Explorative: In the explorative setting, agents behave heavily exploratory during training, thus they must consider the risk brought on by heavily exploration of other agents. In this setting, the annealing time is change to 500K following the setting of QMIX [5].

2. Dilemmatic: In the dilemmatic setting, agents should consider risk to prevent the learning of locally optimal policies. In this setting, agent could receive negative rewards which is based on damage dealt to our agents.

3. Random: In this setting, one agent performs random actions 50% of the time.

As depicted in Figure 8, RiskQ obtains the best performance for the explorative, dilematic, and random 3s_vs_5z. It obtains the best performance for the dilemmatic MMM2. For the explorative MMM2 scenario, although RiskQ learn slowly in the beginning, it obtain the best performance in the end. Combining previous results from the MACN and the MACF environments, we can conclude that RiskQ can yield promising results in environments that require risk-sensitive cooperation.

In the standard SMAC scenarios, we compared the performance of RiskQ with various baseline algorithms. The StarCraft II version used in this study is SC2.4.6, which is consistent with QMIX [5], WQMIX [15], and ResQ [8]. We presented results of six SMAC scenarios (MMM2, MMM,

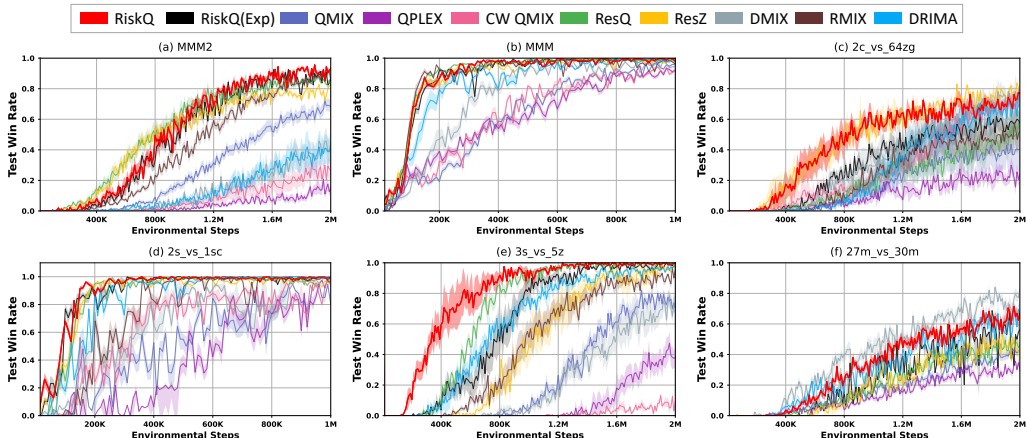

Figure 9: The test win rate of different algorithms (including RiskQ(Exp) which acts based on the expectation of the value distribution) in SMAC scenarios.

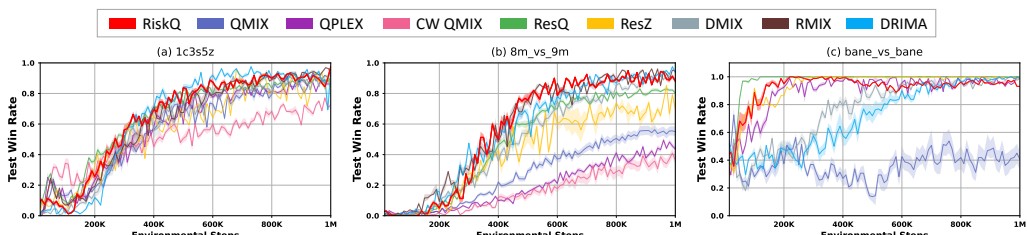

Figure 10: The test win rate of different algorithms for the SMAC benchmark.

2c_vs_64zg, 2s_vs_1sc, 3s_vs_5z, 27m_vs_30m) in Figure 9, revealing that not only can RiskQ achieve state-of-the-art performance in these scenarios, but RiskQ(Expectation) also exhibits decent performance. *The gap between RiskQ and RiskQ(Expectation) underscores the importance of considering risk*. Here we provide additional results for three additional scenarios in Figure 10. As depicted in Figure 10, RiskQ achieves competitive performance in the 1c3s5z, 8m_vs_9m, and bane_vs_bane scenarios.

Additionally, we compared the performance between RiskQ and DRIMA that configured with cooperative risk-seeking and environmental risk-seeking settings. The experimental results are illustrated in Figure 12 (left). The result reveals that RiskQ(Expectation) can achieve performance on par with DRIMA and DRIMA-SS. The consideration of risk in RiskQ allows for further performance enhancement, thereby emphasizing the superiority of RiskQ's performance and the importance of risk consideration.

Moreover, since the optimization metrics of RiskQ are not expectations, it is not fair to compare only the expectation metrics with other baselines. We compare the Wang 0.75 metric for return distribution and win rate distribution in two standard SMAC scenarios. As IQN [20] does not guarantee to converge to the true distribution, we want to know whether the algorithm is optimizing the risk metric correctly. The experimental results in Figure 11 indicate that the risk-sensitive objective optimized by RiskQ is learning gradually with time.

All the SMAC experiments were conducted using the SC2.4.6 version of StarCraft II. However, as noted in WQMIX, *"performance is not comparable across versions"*. Therefore, we conducted experiments using the SC2.4.10 version of StarCraft II to compare the performance of RiskQ with ResQ, ResZ, and fine-tuned QMIX [28] in the MMM2 scenario. As depicted in Figure 12 (right), RiskQ maintains strong performance even in the SC2.4.10 version. This suggest that RiskQ can obtain promising results in SC2.4.10 as well.

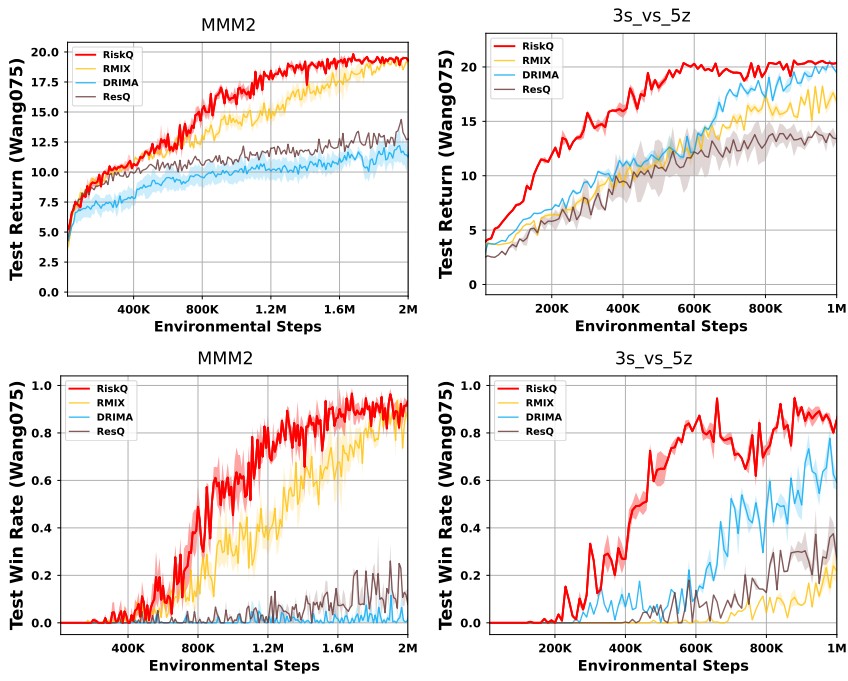

Figure 11: The Wang 0.75 metric for return distribution and win rate distribution in standard SMAC: MMM2 (Left) and 3s_vs_5z (Right).

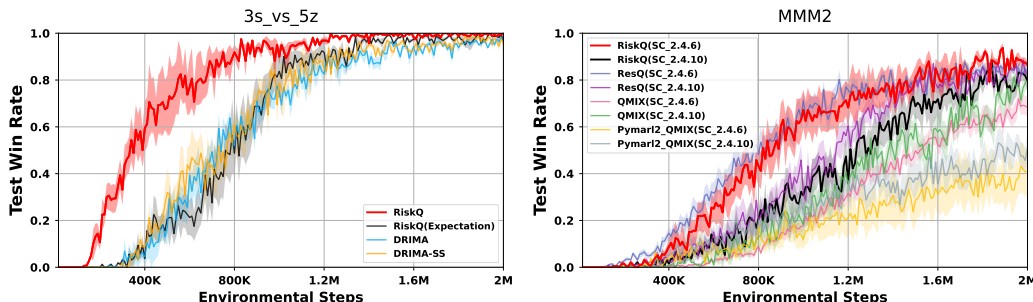

Figure 12: A performance comparison between RiskQ and DRIMA with different configurations under the 3s_vs_5z scenario, and the performance of algorithms with different StarCraft II versions.

## C.5 SMACv2

SMACv2 [54] is a new version of SMAC with improved stochasticity, which can better demonstrate RiskQ's ability to handle uncertainty. We verified the performance of RiskQ against other baselines in different SMACv2 scenarios, and the results are shown in Figure 13.

## C.6 Ablation Study

### C.6.1 Impact of not satisfying the RIGM principle

To investigate the reasons behind RiskQ's promising results, we analyze different designs of RiskQ on the SMAC, MACN, and MACF scenarios. First, we study the necessity of satisfying the RIGM principle by making about 50% of RiskQ agents follow different risk measures. In Figure 14, w/o RIGM $VaR_1$, w/o RIGM $CVaR_1$ and w/o RIGM $CVaR_{0.2}$ indicate that about 50% of agents act according to the $VaR_1$ (risk-seeking), $CVaR_1$ (risk-neutral) and the $CVaR_{0.2}$ (risk-averse) metrics, respectively. These risk measures are not the risk measure $\psi_\alpha = Wang_{0.75}$ used by other agents. As depicted in Figure 14 (a-c) and (e-f), RiskQ performs poorly in all the three cases, highlighting the importance of satisfying the RIGM principle that agents act according to the same risk measure. The performance of these three cases is comparable to that of RiskQ in the 2s_vs_1sc scenario. We

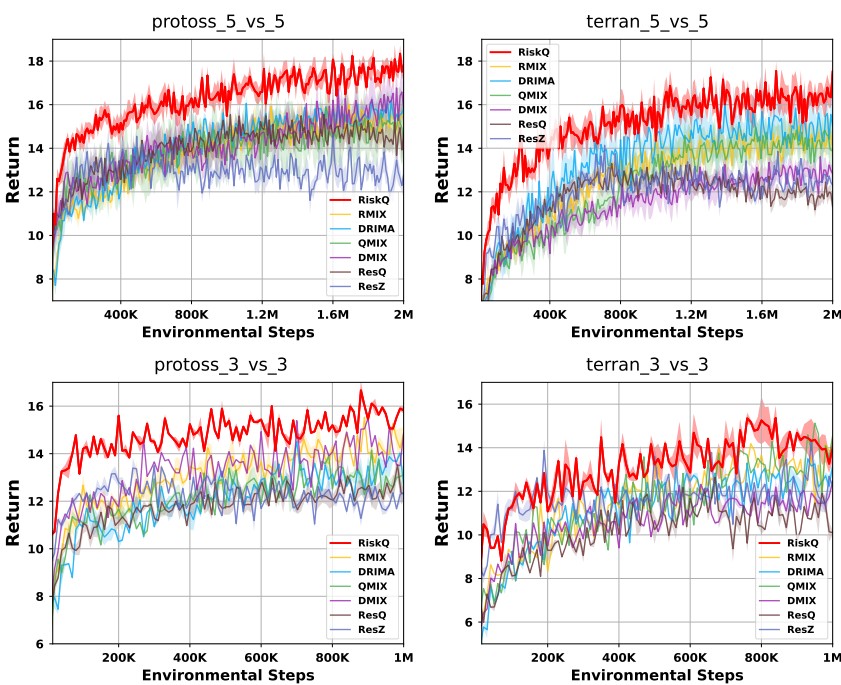

Figure 13: The test return mean of different algorithms in different SMACv2 scenarios.

speculate that this may be due to the fact that 2s_vs_1sc is a simple environment with fewer agents, and therefore, the differences in performance are not apparent.

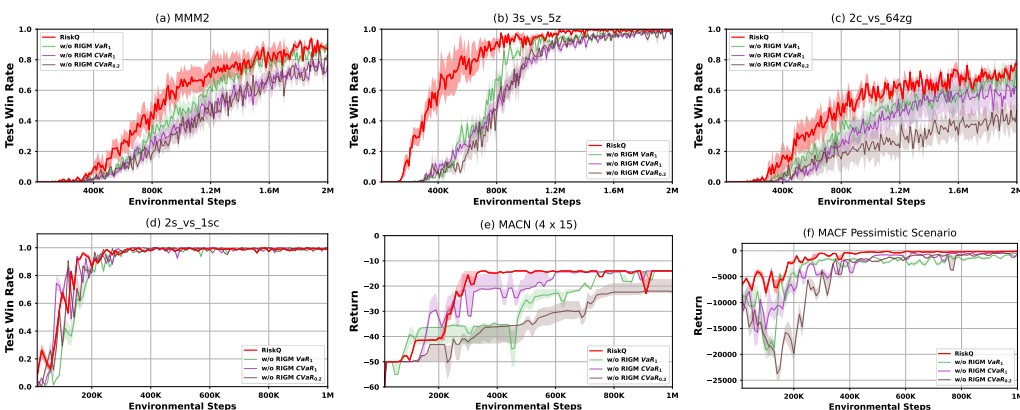

Figure 14: Ablation Study on why the RIGM principle matters.

As shown in Fig 14, not satisfying RIGM could lead to significant performance drop. We will further explain the usefulness of RIGM further through an example and corresponding empirical results.

In cooperative MARL, the optimal action of an agent depends on the actions executed by others. As an example for MARL combat scenarios, depicted in Fig 15, if an agent has doubts about its teammates and leans towards a pessimistic outlook, it may evade rather than confront the enemies. However, with RIGM, all the agents can embrace a risk-seeking strategy. This promotes the agent to adopt a more optimistic perspective on its teammates, which in turn promotes enhanced collaboration. Hence, the incorporation of RIGM is crucial. The DIGM principle can not be used for such scenarios which require risk-sensitive policies.

To empirically study the example, we consider the following two non-RIGM cases on the 3s_vs_5z map of the SMAC benchmark.

1. Each agent acts according to different risk-metrics for their actions. The joint optimal action is derived from each agent's individual risk-based optimal action, representing a non-RIGM approach. Specifically, we assign different risk-metrics to each agent: VaR 1.0, CVaR 1.0, and Wang 0.75, corresponding to risk-seeking, risk-neutral and risk-averse policies, respectively.

2. In the second case, each agent uses the same risk metric as before. Different the first case, when determining the approximated optimal joint action, we uniformly apply Wang 0.75 as the risk metric.

The empirical results are shown in Fig 15, 'Case 1' and 'Case 2' correspond to the first and the second cases, respectively. The results indicate that not satisfying RIGM could lead to performance drop.

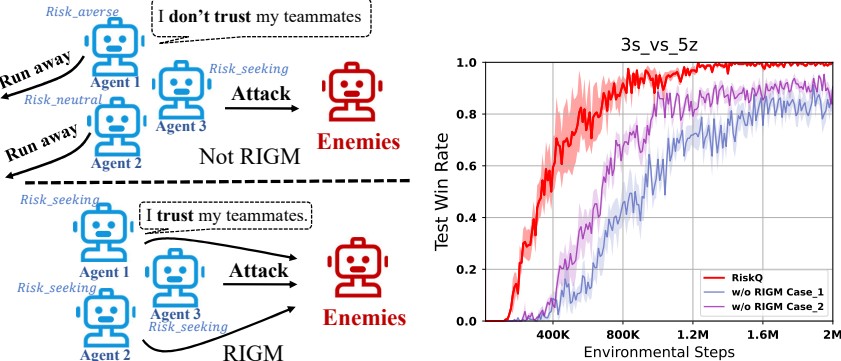

Figure 15: Without RIGM, agents might choose to run away (the upper part of Fig(Left)). With RIGM, agents act in a consistent way. (the lower part of Fig(Left))). Fig(Right): RiskQ performs significantly better than two cases that do not satisfy the RIGM principle.

### C.6.2 Impact of representation limitations

By modeling the percentile of joint return distribution $\theta(\omega)$ as the weighted sum of the percentiles $\theta_i(\omega)$ of each agent's distribution, RiskQ suffers from representation limitations. To study whether the representation limitations impact the performance of RiskQ, we design three variants of RiskQ: RiskQ-QMIX, RiskQ-Residual, and RiskQ-Residual-QMIX.

1. RiskQ-QMIX (Sec. B.3): RiskQ-QMIX models the monotonic relations among $\theta_i(\omega)$ and $\theta(\omega)$ in the manner of QMIX [5]. We study its performance with respect to two risk metrics: $\text{VaR}_{0.2}$ and $\text{VaR}_{0.6}$.

2. RiskQ-Residual (Sec. B.4): RiskQ-Residual models the joint value distribution without representation limitations by using residual functions [8]. We study its performance for the risk metric $\text{Wang}_{0.75}$.

3. RiskQ-Residual-QMIX(Sec. B.5): RiskQ-Residual-QMIX combines RiskQ-Residual and QMIX. We study its performance with respect to two risk metrics: $\text{VaR}_{0.2}$ and $\text{VaR}_{0.6}$

RiskQ-QMIX and RiskQ-Residual-QMIX satisfy the RIGM principle for the VaR metric, and RiskQ-Residual satisfies the RIGM for the VaR and DRM metrics. As it is shown in Figure 16, albeit these variants have better representation ability, their performance is unsatisfied for the MMM2, 3s_vs_5z, 2s_vs_1sc, and 2c_vs_64zg. This suggest that the representation limitation does not significantly impact the performance of RiskQ.

### C.6.3 Impact of different designs of RiskQ mixers

Moreover, in Figure 17, we analyze different designs of the RiskQ mixer through two variants: RiskQ-Sum and RiskQ-Qi.

1. RiskQ-Sum: RiskQ-Sum models the percentile $\theta(\boldsymbol{\tau}, \boldsymbol{u}, \omega)$ as the sum of the percentiles of per-agent's utilities $\sum_{i=1}^{n} \theta_i(\tau_i, u_i, \omega)$.

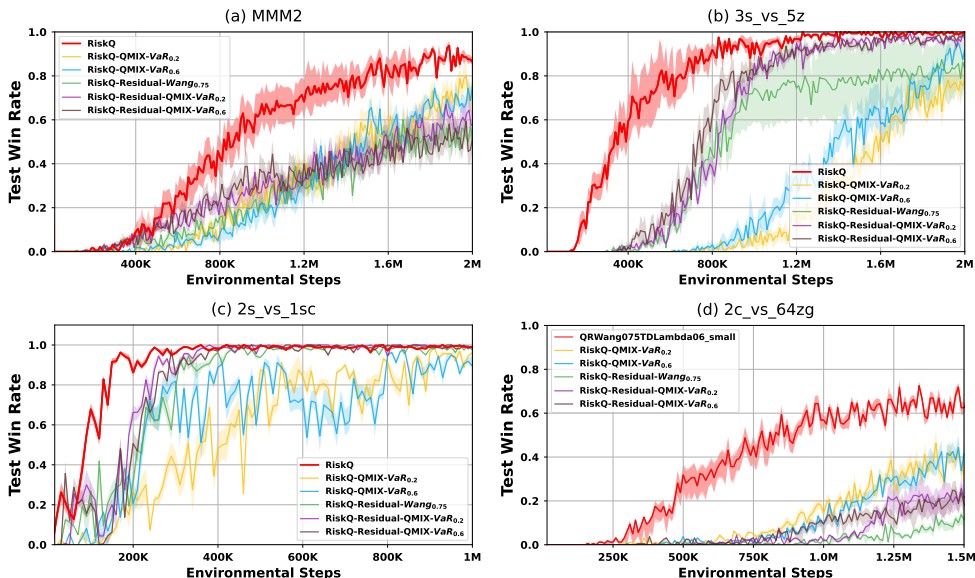

Figure 16: Study on whether the representation limitations impact the performance of RiskQ.

2. RiskQ-Qi: RiskQ-Qi represents each percentile $\theta(\boldsymbol{\tau}, \boldsymbol{u}, \omega)$ as the expectation of state-action function.

As shown in Figure 17, both two variants perform inferior to RiskQ in most cases.

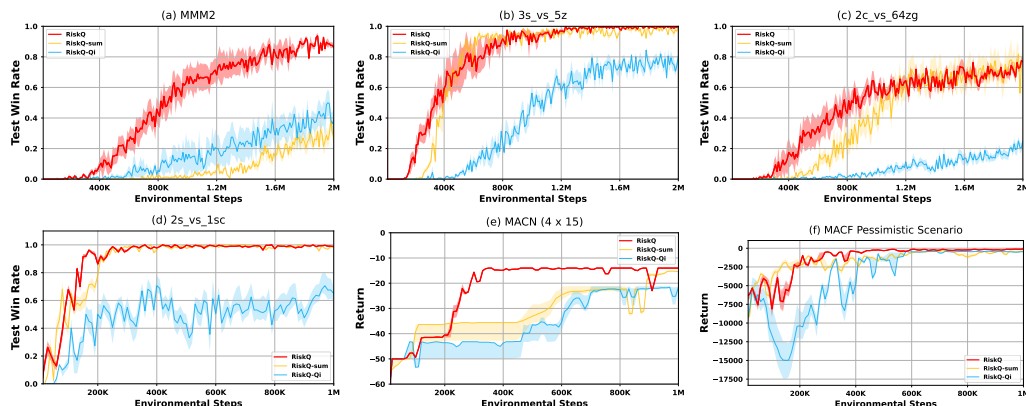

Figure 17: Analysis of different designs of RiskQ mixer.

### C.6.4  Impact of different utility functions

IQN [20] and QR-DQN [19] suffer from the crossing quantile issues that a low quantile $VaR_{0.2}$ could be larger than a high quantile $VaR_{0.5}$ due to its limitation. To address this limitations in MARL, we explore the following variants.

1. QR (sorted): This method models the distribution by fixing the quantiles in the style of QR-DQN [19], and then sorts the quantiles to avoid the quantile-crossing issues.

2. QR (unsorted): This method models the distribution by fixing the quantiles in the style of QR-DQN [19], but does not sort the quantiles.

3. IQN (sorted): This method models the distribution by sampling the quantiles in the way of IQN [20], and sorts the quantiles to ensure a monotonic increasing order of the quantiles.

4. IQN (unsorted): This method models the distribution by sampling the quantiles in the way of IQN [20], but does not sort the quantiles.

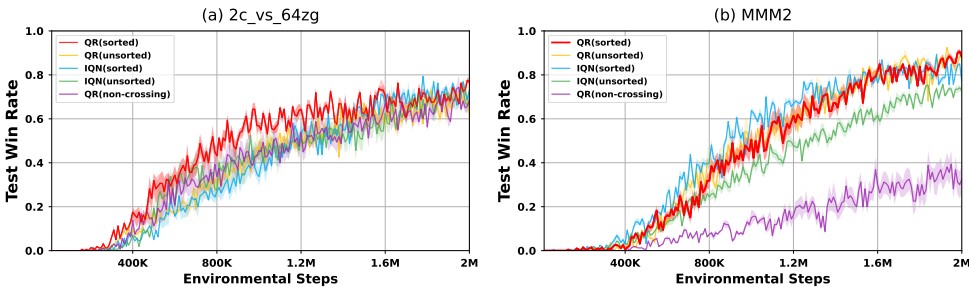

Figure 18: Analysis of different implementations of distribution utility.

5. QR (non-crossing): This method adopts the non-crossing quantile regression to model the distribution as proposed in [55].

As illustrated in Figure 18, we explored several different implementations of distribution utility. QR (sorted) has promising results. So we implement the utility function of RiskQ using QR (sorted).

# D    Discussion

## D.1    Is risk-sensitive RL related to safe RL?

Risk management within RL can be roughly divided into two categories: safe and constrained RL, and distributional risk-sensitive RL. Safe and constraint RL formulates the risk as some kind of constraints within the policy optimization problem. Safe RL is often modeled as a Constrained Markov Decision Process (CMDP)[56], in which the target is to maximize the agent reward while making agents satisfy safety constraints. For instance, [57] proposes a Lagrangian method that provides a theoretical bound on cost function while optimizing the policy; [58] employs the Lyapunov approach to systematically convert dynamic programming and RL algorithms into their safe versions.

However, in complicated real-world applications, the form of risk may either be too complex or the constraints are hard to be explicitly formulated, safe RL algorithms can be challenging to learn. In that case, distributional RL provides a way to utilize risk measures upon the return distributions for risk-sensitive learning. IQN[20] considers the risk as the distortion risk measures upon sampling distributions to realize risk-sensitive policies. [59] learns a full return distribution for its critic and optimizes the actor's policy according to a risk-related metric (such as $CVaR$) for offline RL.

## D.2    Which behaviors in standard SMAC scenarios can potentially involve risk?

Risk refers to the uncertainty in MARL. It is not refer to events that will incur negative outcomes. In the SMAC scenarios, when an agent performs an action, there are three possible sources of risk: observation uncertainty, environmental uncertainty, and agent policy uncertainty. In standard SMAC, as each agent can only access partial observation, their policies contain more randomness than policies with access to the whole state. Environmental uncertainty is caused by the stochasticity of state transitions and the reward function. Agent policy uncertainty comes from the learning process of agents and their exploratory behaviors (e.g., epsilon-greedy). In the standard SMAC, we find that employing a risk-averse policy (Wang 0.75) can lead to better performance. This could be attributed to the reason that the risk-averse policy incorporates a strategy to stay alive longer; this is correlated with the game's objective of eliminating all opponents while staying alive.

In a standard game environment, it is uncommon that using risk-sensitive policies can lead to better performance than risk-neutral policies. For example, IQN[20] finds that risk-averse policies can obtain better results than risk-neutral counterparts in Atari game environments (e.g., ASTERIX and ASSAULT). Furthermore, they also find that risk-seeking policies can obtain better performance in 3 out of 6 standard games.

### D.3  Can RiskQ make the learned CDF function converge to the real distribution?

RiskQ uses implicit quantile network (IQN) [20] as the basis to learn value distribution, so it shares the same converge property as IQN.

As stated in Chapter 7.5 and 7.6 of [50] and Theorem 4.3 of [51], the greedy distributional Bellman update operator of IQN is not a contraction mapping. This is an inherent drawback of the distributional RL. However, the approximation error of some risk metrics can be made arbitrary small by increasing the number of statistics (Theorem 4.7 of [51]) for the policy evaluation case.

As we know, Lim and Malik [52] proposes a modification to IQN with a new distributional Bellman operator recently. They show that the optimal CVaR policy corresponds to a fixed point of the new operator. However, it is unclear whether it converges in general settings. We have combined RiskQ with [52] by replacing IQN with it. As shown in Figure 19, the new method, LimAndMalki, performs poorly. This suggests that remedying the non-contraction mapping issues may not be important enough for performance improvement as existing risk-sensitive RL methods (e.g., IQN) are already working well.

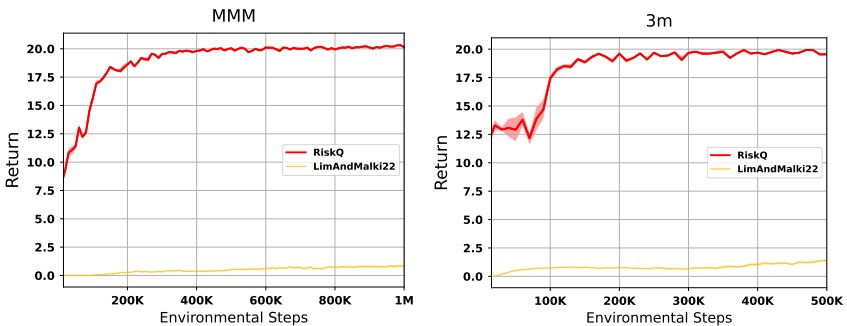

Figure 19: Replacing the IQN (Dabney et al. ICML 2018) used by RiskQ with a risk-sensitive algorithm (Lim and Malik NeurIPS 22) whose optimal CVaR policy corresponds to a fixed point of it.

### D.4  In what way can risk distortion actually be used for exploration-exploitation trade-off?

Likelihood Quantile Networks (LQN) [53] assigns various learning rates for samples (from agent exploration or suboptimality) based on likelihoods, which measures return distribution differences. For exploration, LQN gradually anneals the parameters of its risk-based action selection strategy with time.

We have combined RiskQ with the LQN riks-parameters annealing procedure, and call it as RiskQ+LQN. As is depicted in Fig. 20, RiskQ+LQN performs slightly weaker than RiskQ. This suggests that besides considering the uncertainty for exploration, coordinated exploration may be needed as well. It also provides a new direction for future work.

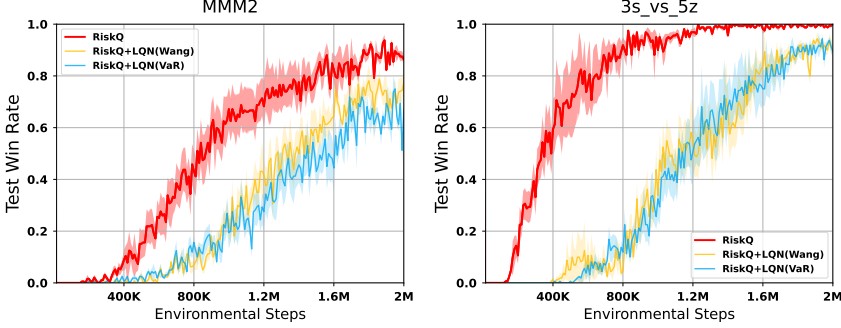

Figure 20: Combining RiskQ with the risk-parameter annealing procedure of likelihood quantile network (Lyu et al. 2019): MMM2 (Left) and 3s_vs_5z (Right). RiskQ+LQN(Wang) uses the Wang metric and anneals the risk parameter from 0.75 to -0.75. RiskQ+LQN(VaR) uses the VaR metric, and the risk parameter gradually anneals from 0.8 to 0.1.

