# OpenReview forum: "RiskQ: Risk-sensitive Multi-Agent Reinforcement Learning Value Factorization"
_NeurIPS.cc/2023/Conference — NeurIPS 2023 poster_

### Official Review · Reviewer_EN1R · 2023-06-09

**Soundness:** 3 good
**Presentation:** 3 good
**Contribution:** 3 good
**Rating:** 6
**Confidence:** 2

**Summary:**

This paper studies risk-sensitive action selection in cooperative multiagent reinforcement learning (MARL) under the centralized training but decentralized execution (CTDE) paradigm, based on the value decomposition class of methods. The principle underlying value decomposition is Individual Global Max (IGM), with its variant under distributional learning being called Distributional IGM (DIGM). This paper generalizes IGM and DIGM for risk-sensitive behavior, resulting in the Risk-sensitive IGM (RIGM). The sources resulting in risk are environmental uncertainty (stochasticity in state dynamics and/or rewards), non-stationary policies of agents (all agents are learning/evolving), and partial observability (perceptual ambiguity: observations have missing information in comparison to the Markov states).

Given that the problem setting is cooperative MARL, the notion of risk is one that minimizes risk on the global or collective action, and not individual consequences per agent.

**Strengths:**

- The contribution is novel (as far as I know)

- Problem is well motivated and exposition is done sufficiently well

- Experiments are thorough, with a good collection of environments/tasks and key baselines included

**Weaknesses:**

Limitations are not thoroughly discussed (my opinion could change subject to authors' response)

**Questions:**

- Are there any none value decomposition baselines for dealing with risk for CTDE?

- Isn't there a simpler way to achieve risk-aversive sub-policies by having a risk-aversive central critic update (as in Dabney et al. (2018b))? Maybe the issue there is not being able to ensure individual policies are also risk aversive?

- What are the limitations of this approach and potential future works?

**Limitations:**

Not discussed in good detail (my opinion could change subject to authors' response)

---

> ### Author Rebuttal · Authors · 2023-08-06
>
> We want to express our sincere gratitude to the reviewer. Such a thoughtful and in-depth review can help us greatly improve the quality of this work. We hope that with our response, the reviewer can kindly reconsider the final opinion. We answer the comments as follows.
>
> **Limitations are not thoroughly discussed (my opinion could change subject to authors' response)**
>
> Due to space limitations, please refer to the global response for a thorough discussion of the limitations of our work.
>
> **Are there any none value decomposition baselines for dealing with risk for CTDE?**
>
> To systematically answer the reviewer's question, we have conducted a comprehensive survey of accepted papers pertaining to *risk* across several major conferences: NeurIPS (2019-2022), ICML (2019-2023), ICLR (2019-2023), AAMAS (2019-2023), and AAAI (2019-2022). For each conference, the number of papers that contain *risk* in its title is listed as follows.
>
> |  | NeurIPS |ICML| ICLR| AAAI|AAMAS|
> |--|--|--|--|--|--|
> |2019|9| 8|1| 6|1|
> |2020|12|3|0|9|1
> |2021|24|16|7|15|4
> |2022|19|22|27|11|1
> |2023|N/A|15|25|8|4
>
> Through our detailed investigation, there are a total of 248 papers have *risk* in its title from these conferences. Besides, we also search all the papers that cite QMIX, a popular value factorization method. For the 1170 papers that cite QMIX， there are 7 papers contains risk in its title. Then, for those 255 papers, we read its paper title and abstract to determine whether it is related to both risk and MARL. Only 24 out of the 255 papers consider the risk factor in MARL. For these 24 papers, we find that there does not exist any non-value-decomposition method dealing with risk for CTDE.
>
>
>
>
>
> **Achieve risk-aversive sub-policies by having a risk-aversive central critic update**
>
>
> Having a risk-averse central critic cannot ensure that individual policies are also risk-averse. In fact, a similar idea has been explored in DRIMA [1]. In DRIMA, the central critic can be updated with various risk-sensitive policies, while each individual agent can use various risk-sensitive policies to select actions. We have shown in Theorem 3 that DRIMA cannot guarantee adherence to the RIGM principle for the CVaR metric. Moreover, such an idea could lead to a performance drop, as we have shown in our work that RiskQ performs better than DRIMA in all the risk-sensitive scenarios and all the risk-neutral scenarios.
>
> To further answer the reviewer's question, we explore the idea of updating the central critic using a risk-averse policy, and each individual agent acts according to a risk-neutral policy. Through this way, we have designed two new algorithms: simple-RiskQ and simple-ResZ, which are variations of the RiskQ and ResZ [2] algorithms. The experimental results are depicted in Fig. 6 (bottom left) of the response PDF. As it is shown in the figure, these two methods perform infer to RiskQ. The experimental results indicates the importance of satisfying the RIGM principle. Without satisfying the RIGM principle, the algorithm's performance drops.
>
>
> **Potential future works?**
>
> There are three directions worth exploring.
> 1. Risk measures the uncertainty in MARL. A recent paper [3] demonstrates that risk-based exploration can significantly improve the performance of MARL. It would be beneficial to combine RiskQ with uncertainty-based exploration methods. The new algorithm can be risk-seeking for unseen state-action pairs while risk-averse for highly visited state-actions pairs. For environments that require coordinated exploration, it seems that coordination criteria for exploration could be developed, mimicking the RIGM principle. For example, all agents should explore the environment in a coordinated way with the same way of measuring uncertainty. By following such exploration criteria, each agent could explore an environment with high efficiency, which could lead to improved performance.
> 3. RiskQ suffers from representation limitations. However, as is shown in our paper that such limitations do not impact the performance significantly, and it performs even better than its representation-limitation-free variants. It would be interesting to improve the representation-limitation-free variants through advanced neural network designs. We will explore more neural network designs, loss designs, and learning algorithms for these variants.
> 4. As a method that is centrally trained and decentralized executed, RiskQ may suffer from scalability issues for a large number of agents. It would be beneficial to explore the possibility of extending RiskQ to scenarios with independent learners which have better scalability. However, for independent learners, due to the lack of the central state used for training, their learning algorithms may contain more risk than CTDE algorithms. It is interesting and challenging to adapt RiskQ to the independent learner setting.
>
> **Reference**
>
> [1] Kyunghwan Son, Junsu Kim, Sungsoo Ahn, Roben Delos Reyes, Yung Yi, and Jinwoo Shin. Disentangling sources of risk for distributional multi-agent reinforcement learning. ICML 2022.
>
> [2]  Siqi Shen, Mengwei Qiu, Jun Liu, Weiquan Liu, Yongquan Fu, Xinwang Liu, and Cheng Wang. ResQ: A Residual Q function-based approach for multi-agent reinforcement learning value factorization. NeurIPS 2022.
>
> [3] Jihwan Oh, Joonkee Kim, Minchan Jeong, Se-Young Yun, Toward Risk-based Optimistic Exploration for Cooperative Multi-Agent Reinforcement Learning. AAMAS 2023

---

> ### Author Response · Authors · 2023-08-19
>
> Dear Reviewer EN1R,
>
> We have answered the question regarding value decomposition baselines with risk, thoroughly discussed the limitations (in the global response), and discussed future directions. We have studied the performance of having a risk-aversive central critic update; the results are shown in the response PDF.
>
> We are looking forward to your reply.
>
> Please let us know if you have any other concerns.
>
> Thanks and Best Regards,
>
> Authors of Submission 6208

---

> > ### Comment · Reviewer_EN1R · 2023-08-19
> >
> > Thank you for your response which clarifies my concerns. I'm happy to increase my score.

---

> > > ### Author Response · Authors · 2023-08-19
> > >
> > > Thanks for your quick response. We are happy that your concerns have been addressed and appreciate that the rating has increased.

---

### Official Review · Reviewer_rLaN · 2023-06-15

**Soundness:** 3 good
**Presentation:** 3 good
**Contribution:** 3 good
**Rating:** 6
**Confidence:** 1

**Summary:**

The authors introduce the Risk-sensitive Individual-Global-Max (RIGM) principle as a generalization of the Individual-Global-Max (IGM) and Distributional IGM (DIGM) principles and propose RiskQ satisfies the RIGM principle. The method is evaluated by both theoretical analysis and experiments.

**Strengths:**

1. The paper is easy to follow.
2. The idea of considering risk in multi-agent reinforcement learning is sound.
3. The method is evaluated by both theoretical analysis and experiments.

**Weaknesses:**

1. It is possible that considering risk could potentially be detrimental to solving tasks that suffer from the relative overgeneralization problem. Agents may exhibit a tendency to select less risky actions, leading to convergence towards a conservative policy. It would be beneficial if the authors could analyze the representation capacity of their proposed method.

**Questions:**

1. Is this work also related to safe RL?
2. Which behaviors in standard SMAC scenarios can potentially involve risk?

---

> ### Author Rebuttal · Authors · 2023-08-07
>
> We want to thank the reviewer for the effort and time in reviewing this paper. We will take your valuable feedback into account to improve our work. We hope that the reviewer will have a better understanding of our work and a higher confidence in the reviewer's assessment.
>
> ***Representation capacity***
>
> As we had stated in lines 319 and 320 of the submission, RiskQ suffers from representation limitations in that the quantiles of the joint return distribution can be modeled only as a linear combination of per-agent's return distributions. RiskQ does not support more complex functional relationships such as monotonic increasing functions. To address the representation limitations of RiskQ, we had designed three variants of RiskQ: RiskQ-QMIX, RiskQ-Residual, and RiskQ-Residual-QMIX. RiskQ-QMIX can model the monotonic relationships among quantiles. RiskQ-Residual and RiskQ-Residual-QMIX were shown to be free from representation limitations and follow the RIGM principle for some risk metrics. Their details and proofs were provided in B.3, B.4, and B.5 of the Appendix. RiskQ performs better than its representation-limitation-free variants. As shown in Fig. 4 (e) and (f) of the submission and in Section C.5.2 of the appendix, such representation limitations do not negatively impact its performance on the challenging StarCraft II environments. RiskQ performs even better than its three variants which have better representation ability.
>
>
> ***Is this work also related to safe RL?***
>
> Risk management within RL can be roughly divided into two categories: safe and constrained RL, and distributional risk-sensitive RL. Safe and constraint RL formulates the risk as some kind of constraints within the policy optimization problem. Safe RL is often modeled as a Constrained Markov Decision Process (CMDP) [1], in which the target is to maximize the agent reward while making agents satisfy safety constraints. For instance, [2] proposes a Lagrangian method that provides a theoretical bound on cost function while optimizing the policy; [3] employs the Lyapunov approach to systematically convert dynamic programming and RL algorithms into their safe versions.
>
> However, in complicated real-world applications, the form of risk may either be too complex or the constraints are hard to be explicitly formulated, safe-RL algorithms can be challenging to learn. In that case, distributional RL provides a way to utilize risk measures upon the return distributions for risk-sensitive learning. [4] considers the risk as the distortion risk measures upon sampling distributions to realize risk-sensitive policies. [5] learns a full return distribution for its critic and optimizes the actor's policy according to a risk-related metric (such as $\operatorname{CVaR}$) for offline RL.
>
>
> ***Which behaviors in standard SMAC scenarios can potentially involve risk?***
>
> Risk refers to the uncertainty in MARL. It is not refer to events that will incur negative outcomes. In the SMAC scenarios, when an agent performs an action, there are three possible sources of risk: observation uncertainty, environmental uncertainty, and agent policy uncertainty. In standard SMAC, as each agent can only access partial observation, their policies contain more randomness than policies with access to the whole state. Environmental uncertainty is caused by the stochasticity of state transitions and the reward function. Agent policy uncertainty comes from the learning process of agents and their exploratory behaviors (e.g., epsilon-greedy). In the standard SMAC, we find that employing a risk-averse policy (such as Wang 0.75) can lead to better performance. This could be attributed to the reason that the risk-averse policy incorporates a strategy for stay alive longer; this is correlated with the game's objective of eliminating all opponents while staying alive.
>
> In a standard game environment, it is uncommon that using risk-sensitive policies can lead to better performance than risk-neutral policies. For example, IQN [4] finds that risk-averse policies can obtain better results than risk-neutral counterparts in Atari game environments (e.g., ASTERIX and ASSAULT). Furthermore, they also find that risk-seeking policies can obtain better performance in 3 out of 6 standard games.
>
>
> **References**
>
> [1] Eitan Altman. Constrained Markov decision processes. CRC Press, 1999.
>
> [2] Joshua Achiam, David Held, Aviv Tamar, and Pieter Abbeel. Constrained policy optimization. ICML 2017.
>
> [3] Yinlam Chow, Ofir Nachum, Edgar Duenez-Guzman, and Mohammad Ghavamzadeh. A lyapunov-based approach to safe reinforcement learning. NeurIPS 2018.
>
> [4] Will Dabney, Georg Ostrovski, David Silver, and Rémi Munos. Implicit quantile networks for distributional reinforcement learning. ICML 2018.
>
> [5] Núria Armengol Urpí, Sebastian Curi, and Andreas Krause. Risk-averse offline reinforcement learning. ICLR 2021.

---

> > ### Comment · Reviewer_rLaN · 2023-08-15
> >
> > Thanks for the response.

---

> > > ### Author Response · Authors · 2023-08-16
> > >
> > > Thanks for the reviewer's time and effort to read our response. We are eager to discuss with the reviewer regarding our work. Here, we describe our work further to give the reviewer a better overview of the value factorization methods. Our work is a value factorization approach. Value factorization (or value decomposition) methods learn a factored state-action value function (utility function) for each agent. Each agent acts greedily according to the state-acton function.
> > >
> > > For value factorization methods, the **individual-global-max (IGM)** principle is an important principle. It requires that the optimal joint action equal to the collection of individual optimal actions. Many approaches have been proposed. For example, VDN [1] models the joint state-action value function as the sum of each individual state-action value function. QMIX [2] models the joint state-action value function as a monotonic increasing function with respect to each individual state-action value function. QTran [3] can factorize any factorized state-action value function and does not suffer from representation limitations. However, it is sample inefficient. QPLEX [4] and WQMIX [5] use different methods to learn state-action value functions that do not suffer from representation limitations.
> > >
> > > DMIX [6] extends the individual-global-max (IGM) principle to the **distributional individual-global-max (DIGM) principle**, which is used for distributional MARL. DIGM principle requires that for distributional state-action value function (stochastic value function), the optimal joint action is equal to the collection of individual optimal actions. However, it suffers from representation limitations too. ResZ [7] uses a residual function-based approach, it satisfies the DIGM principle without representation limitations.
> > >
> > > In this work, we believe that for risk-sensitive MARL scenarios, to learn factored risk-sensitive state-action value functions for each agent, a new IGM principle should be developed. The **risk-sensitive individual-global-max (RIGM)** principle requires that for risk-sensitive MARL, the optimal risk-sensitive joint action must equal the collection of individual risk-sensitive optimal actions.
> > >
> > > In this work, we use distributional RL algorithms and use risk-metric (e.g., CVaR) to calculate the risk from the return distribution of a state-action pair. It is non-trivial to design an algorithm that satisfies the RIGM principle. As we show in Theorem 1, methods that satisfy the IGM principle **do not guarantee adhere to the RIGM principle**, and in Theorem 2, methods that satisfy the DIGM principle **do not guarantee adhere to the RIGM principle** for some risk-metrics. Further, we have shown that the risk-sensitive MARL algorithm RMIX [8] satisfies the RIGM principle only for a special case, and the risk-sensitive MARL algorithm DRIMA [9] does not satisfy the RIGM principle for the CVaR metric.
> > >
> > > We have developed RiskQ, which satisfies the RIGM principle for the VaR and distorted risk measures. In the paper and the response PDF, we have shown that RiskQ performs significantly better than the other algorithms in risk-sensitive MARL scenarios. Through ablation studies, we show that it is not satisfying the RIGM principle will lead to a performance drop. And RiskQ performs better than its representation-limitation-free variants.
> > >
> > > **Reference**
> > >
> > > [1] Sunehag et al. Value-decomposition networks for cooperative multi-agent learning based on
> > > team reward. AAMAS 2018.
> > >
> > > [2] Rashid et al. QMIX: monotonic value function factorisation for deep multi-agent reinforcement learning. ICML 2018.
> > >
> > > [3] Son et al. QTRAN: learning to factorize with transformation for cooperative multi-agent reinforcement learning. ICML 2019.
> > >
> > > [4] Wang et al. QPlex: Duplex dueling multi-agent q-learning. ICLR, 2021.
> > >
> > > [5] Rashid et al. Weighted QMIX: expanding monotonic value function factorisation for deep multi-agent reinforcement learning. NeurIPS, 2020.
> > >
> > > [6] Sun et al. DFAC framework: Factorizing the value function via quantile mixture for multi-agent distributional q-learning. ICML, 2021.
> > >
> > > [7] Shen et al. ResQ: A residual q function-based approach for multi-agent reinforcement learning value factorization. NeurIPS 2022.
> > >
> > > [8] Qiu et al. RMIX: learning risk-sensitive policies for cooperative reinforcement learning agents. NeurIPS, 2021.
> > >
> > > [9] Son et al. Disentangling sources of risk for distributional multi-agent reinforcement learning. In ICML, 2022.

---

### Official Review · Reviewer_X7fQ · 2023-07-06

**Soundness:** 2 fair
**Presentation:** 3 good
**Contribution:** 2 fair
**Rating:** 5
**Confidence:** 3

**Summary:**

The authors propose RiskQ, a value function factorization method for MARL that considers the risk factor. Risk is a concept present in reinforcement learning, and the authors propose a generalisation of the two popular principles IGM and DIGM: the RIGM, that considers the factor of risk in MARL.


**Strengths:**

In general, the content of the paper is well structured and organized.

- Provides extensive theoretical analysis of the proposed method
- Extends current popular principles in MARL: IGM and DIGM
- Provides empirical results in a wide range of different environments



**Weaknesses:**

- In section 2, the authors mention Dec-Pomdps bu they do not introduce it. Additionally, none of the notation is defined, making it harder to understand some of the concepts, although these can be understood. This should be improved for clarity and completeness purposes.
- In definition 3, value at risk is defined and is stated before as being a common metric (stated in the abstract, for example). Yet, I cannot find a reference for this metric in the text when value at risk is mentioned.
- In definition 4, the DRM is defined, but it is unclear for me what $g(\omega)$ exactly represents; this is not mentioned.
- In line 161 it is stated that the RIGM principle is a generalization of DIGM and IGM. I believe it would be more accurate to say that RIGM is a generalization of DIGM since this is already a generalization of the IGM.
- In section 5.4 the authors describe some scenarios that are risk-averse versions of SMAC introduced in other works. For completeness, the descriptions and specifications of these environments should also be part of this paper.
- In line 33: "Risk refers to the uncertainty of future outcomes in multi-agent systems". I can see some parallelisms between this and curiosity-based approaches, that aim to improve exploration based on states that they don't know. In fact, from my understanding, risk is also a way to improve exploration. I believe it would be beneficial to include some discussions with this or other exploration methods.

Minor:
- line 80: "They models"
- line 59 and line 61: "jointed"
- in line 102-103: "In this work, we interchangeablt use the terms: stochastic value function and return distribution" but right ahead in lines 133-134 the authors distance again the two terms "global value function (rather than a value distribution)"; these ambiguities can be confusing for the reader

Although it seems promising, I believe that this paper still contains some lack of clarity that needs to be improved.

Please find some questions ahead.

**Questions:**

- In section 4.3, in lines 231-232 it is stated how $\psi$ is integrated in the learning problem. According to line 158, $\psi_a$ is expected to return a reward. How is this integrated in the learning problem? It is unclear to me.
- According to section 4.3 and more precisely lines 236-240, I don't see hoe the risk factor is also applied on $Z_{jt}$ as stated in Definition 5 (RIGM). From the description in these lines it sounds to be like RMIX, as described in line 192. Could the authors clarify this?

**Limitations:**

Please find above.

---

> ### Author Rebuttal · Authors · 2023-08-07
>
> We thank the reviewer for your time and effort in reviewing this submission. We will describe the definition of DEC-POMDPs, the VaR metric, the DRM, the risk-averse SMAC, and the risk-sensitive action selection in more detail and include them in the revised paper. The source code of RiskQ is included in the supplementary material. We hope that with the major concern (the clarity issues) resolved, the reviewer can kindly reconsider the final ratings. We answer the comments as follows.
>
> 1. The description of DEC-POMDPs and notations were provided in Appendix A.1.
>
> 2. In line 57, it was written as "VaR (i.e., percentile)". Percentiles are a type of quantiles. The math form of the Value at Risk (VaR) metric was provided in line 107. We will rewrite the formula as  $\operatorname{VaR}_\alpha(Z(\mathbf{\tau}, \mathbf{u}))=\theta(\mathbf{\tau}, \mathbf{u}, \alpha)$. The reference for VaR is "Linsmeier and Pearson. Value at Risk, Financial Analysts Journal, 2000".
>
> 3. We had provided details of the DRM in Appendix A.2. The distortion function is a non-decreasing function $g:[0,1] \rightarrow[0,1]$ satisfying $g(0)=0$ and $g(1)=1$. The distorted expectation of $Z$ is distorted by $g^{\prime}(\omega)$.
>
> 4. It would be more accurate to say RIGM is a generalization of DIGM.
>
> 5. Due to space limitations, the risk-averse versions of SMAC were described in lines 282-284 and its detailed descriptions were provided in lines 837-846 in Appendix C.4.
>
> 7. Exploration [1] in RL can be categorized into two major categories: uncertainty-oriented exploration and intrinsic motivation-oriented exploration. The uncertainty-oriented exploration method makes use of epistemic and aleatoric uncertainty to guide exploration. Intrinsic-oriented exploration makes use of various curiosity information as the intrinsic motivation of exploration. Uncertainty-based methods show general improvements in exploration in most environments, and the intrinsic-based method is more suitable for environments with sparse rewards. We will include a discussion of a few related methods that used distributional RL for exploration in our paper.
>
>     Risk metrics can be used for exploration. NDQFN [2] uses the distributional prediction error (DPE) as the intrinsic reward $i(s,a)$. An agent is encouraged with high intrinsic reward to explore state-action pairs with high DPE. They select action according to $a^* = \arg \max_{a'}(Q(s,a')+c_ti(s,a))$. In [3], $i(s,a)$ is calculated as $\sqrt{\sigma^2_+}$, where $\sigma_+$ is a risk-metric calulated as the left truncated variance of a value distribution.
>
>     We had cited a risk-based MARL exploration work [4] in line 132. [4] and Likelihood Quantile Networks (LQN) [5] select actions according to $\arg \max_{a'}\psi_{\alpha}[Z_(s,a)]$, where $\psi_{\alpha}$ is a risk metric, $\alpha$ is the risk-related parameter and it anneals with time. Agents could be risk-seeking in the beginning and thus has good exploration ability and could be risk-averse at the end of training for exploitation.
>
>
> **Minors**
>
> We will fix typos accordingly and fix all the typos through careful inspection.
>
> The output of a value function is a single scalar value. The output of a stochastic value function (also called value distribution) is a set of quantile values. In 133-134, we want to emphasize that RMIX does not learn a value distribution. And "global" means that the learned value function is $Q_{jt}$, it is not the individual value function $Q_i$.
>
> **Questions**
>
> **How the risk-metric is integrated in the learning problem**
>
> In Sec.B.6.2 of the appendix, the detailed RiskQ learning algorithm and the framework of RiskQ were presented. In lines 6-7 of the algorithm, an agent obtains value distribution $Z(\tau_i, u_i)$ from its agent network. Then, it selects actions according to $\bar{u_i}=\arg \max_u \psi_\alpha[Z(\tau_i, u_i)]$. In line 13 of the algorithm, quantiles of $Z_{jt}$ are calculated as mixture of quantiles of $Z_i$ according to (10). It is computational intractable to find the optimal action $\bar{\mathbf{u}}$ for $\psi_\alpha[Z_{jt}(\tau, u)]$. For practical implementation, we use an approximation to obtain the optimal action, following popular value factorization methods (e.g., QMIX). The optimal joint action $\bar{\mathbf{u}}$ is approximated by $[\bar{u_i}]_{i=1}^N$. Then, the quantile loss is calculated according to lines 236-240.
>
>
> **How the risk-metric is applied for $Z_{jt}$**
>
> The process of applying risk factor on $Z_{jt}$ is exactly the same as the process for applying it on $Z_i$. For a given state-action pair $(\tau, u)$, the neural network implementing $Z_{jt}$ outputs a set of values $\theta(\tau, u, \omega_j)$, representing the quantile value at $\omega_j$. Taking the risk metric $\psi_\alpha = VaR_{0.2}$ as an example, $\psi_{\alpha}[Z_{jt}]$ will output $\theta(\tau, u, 0.2)$.
>
> There are 3 major differences between RMIX and RiskQ. 1) RMIX learns a joint value function $Q_{jt}$, which is a scalar value rather than a distribution. RiskQ learns $Z_{jt}$ as a value distribution. 2)The optimal joint action of RMIX is selected according to the single expected value of $Q_{jt}$, while RiskQ selects the optimal action according to $\psi_\alpha[Z_{jt}]$. 3) RMIX cannot learn risk-seeking policies whereas RiskQ can.
>
>
>
>
>
>
> **References**
>
> [1] Hao et al. Exploration in Deep Reinforcement Learning: From Single-Agent to Multiagent Domain, TNNLS 2023
>
> [2] Zhou et al. Non-decreasing Quantile Function Network with Efficient Exploration for Distributional Reinforcement Learning. IJCAI 2021
>
> [3] Mavrin et al. Distributional reinforcement learning for efficient exploration. ICML 2019
>
> [4] Oh et al. Risk perspective exploration in distributional reinforcement learning. NeurIPS workshop, 2022
>
> [5] Lyu and Amato. Likelihood Quantile Networks for Coordinating Multi-Agent Reinforcement Learning. AAMAS 2020

---

> > ### Comment · Reviewer_X7fQ · 2023-08-15
> > **Thanks**
> >
> > Thanks for your response. Most of my questions were addressed. However, I believe the paper still needs some improvements in terms of clarity, as there are several terms that are not easy to follow with the flow of the paper. Additionally, there is a lot of important information in the Appendix that is not referred to in the main paper. I suggest that at least the authors refer to the appendix more often in the main body.
> >
> > With this being said, I will raise my score a bit since by main concerns have been answered, but I still believe the paper could be a bit improved in terms of clarity.

---

> > > ### Author Response · Authors · 2023-08-16
> > >
> > > Thanks for your time and effort to read our response. We are happy to see that most of the questions are addressed. We will improve the clarity of this paper. For example, we will describe the two terms return distribution and stochastic value function in a better way. And we will refer to the appendix more often in the revised paper.
> > >
> > > Are there any other questions that we do not address? We are eager to answer and address the drawbacks of this paper. Your opinion is valuable for us to improve the quality of this work.

---

### Official Review · Reviewer_e297 · 2023-07-21

**Soundness:** 3 good
**Presentation:** 3 good
**Contribution:** 2 fair
**Rating:** 6
**Confidence:** 4

**Summary:**

This paper propose a factored distributional value learning for multi-agent reinforcement learning. The paper also propose a theoretically desirable factored risk-distortion property to which the proposed method adhere.

**Strengths:**

The theoretical argument establishes an intuitively sound criterion for evaluating and developing risk distorted methods. The manuscript is written clearly, the theoretical results are proved, and details are comprehensively provided in the appendix.

**Weaknesses:**

The empirical improves are marginal and does not seem to validate the claim of “RiskQ can yield promising results in environments that require risk-sensitive cooperation”, especially considering the quantile loss update includes potentially many quantile samples, effectively increasing the batch-size; not to mention the distributional approach itself provides a richer learning target and potential performance gains.

**Questions:**

RIGM seems very important on a theoretical stand point according to the manuscript, how important is it empirically? Any simple counterexamples that proves its usefulness? or deficiencies of DIGM?

The correctness argument for the proposed method seems to assume the learned CDF, in one way or another, resemble or converge onto the true distribution. How sound is this assumption? Can it be validated in the experiments? If so, it can significantly strengthen the correctness argument.

A related work, likelihood quantile networks (Lyu et al.), mention that risk-seeking may be beneficial in MARL exploration, but obviously it depends. In what way would you recommend risk distortion can actually be used for exploration-exploitation trade-off?

**Limitations:**

The method has the same limitations as other factored value learning methods, such as scalability. In addition, another limitation is that the method does not readily extend to mixed-reward settings in a scalable way.

---

> ### Author Rebuttal · Authors · 2023-08-08
>
> We want to express our sincere gratitude to the reviewer. Such a thoughtful and in-depth review can help us greatly improve the quality of our work.
>
> **The empirical improves are marginal ... require risk-sensitive cooperations**
>
> As shown in Fig. 2 of our work, RiskQ performs ***significantly better*** than other methods in environments which require **risk-sensitive cooperation**. It performs significantly better than all other risk-sensitive methods and popular risk-neutral methods in risk-sensitive SMAC, MACliff, and MACF. The reason why RiskQ obtains only marginal improvement on standard SMAC (in Fig. 3 of our work) is that SMAC does not have enough uncertainty which requires complex risk-sensitive policies[1].
>
> We have evaluated RiskQ in *random* SMAC and SMACv2[1]. In random SMAC, during testing, one agent performs randomly in 50% of the time, which increases collaboration uncertainty. SMACv2[1] is a new version of SMAC with improved stochasticity.
>
> The experimental results are depicted in Fig. 1 of the response PDF. In random SMAC and SMACv2, RiskQ outperforms all other algorithms by a large margin in all scenarios. This demonstrates RiskQ's superiority for environments that require risk-sensitive cooperation.
>
> **the distributional approach ... potential performance gain**
>
> The performance gain of RiskQ does not mainly come from the use of distributional RL. There are 4 distributional value factorization methods: RMIX, DRIMA, DMIX, and ResZ. RiskQ performs better than them by a large margin, as is shown in Fig. 2 of our work. Albeit distributional RL can effectively increase the batch size by using more quantiles, increasing the quantile count does not necessarily improve its performance, as shown in Fig. 4(i) of our work.
>
> **RIGM ... how important is it empirically ... counterexamples**
>
>
> As shown in Fig.4 (a-d) of our work and Sec. 5.1 of the appendix, not satisfying RIGM could lead to significant performance drop. We will further explain the usefulness of RIGM further through an example and corresponding empirical results.
>
> In cooperative MARL, the optimal action of an agent depends on the actions executed by others. As an example for MARL combat scenarios, depicted in Fig. 2 of the response PDF, if an agent has doubts about its teammates and leans towards a pessimistic outlook, it may evade rather than confront the enemies. However, with RIGM, all the agents can embrace a risk-seeking strategy. This promotes the agent to adopt a more optimistic perspective on its teammates, which in turn promotes enhanced collaboration. Hence, the incorporation of RIGM is crucial. The DIGM principle can not be used for such scenarios which require risk-sensitive policies.
>
> To empirically study the example, we consider the following two non-RIGM cases on the 3s_vs_5z map of the SMAC benchmark.
>
> 1. Each agent acts according to different risk-metrics for their actions. The joint optimal action is derived from each agent's individual risk-based optimal action, representing a non-RIGM approach. Specifically, we assign different risk-metrics to each agent: VaR 1.0, CVaR 1.0, and Wang 0.75, corresponding to risk-seeking, risk-neutral and risk-averse policies, respectively.
>
> 2. In the second case, each agent uses the same risk metric as before. Different the first case, when determining the approximated optimal joint action, we uniformly apply Wang 0.75 as the risk metric.
>
>
> The empirical results are shown in Fig. 2 of response PDF, 'Case 1' and 'Case 2' correspond to the first and the second cases, respectively. The results indicate that not satisfying RIGM could lead to performance drop.
>
> **The correctness argument ... How sound is this assumption?**
>
> RiskQ uses implicit quantile network (IQN)[2] as the basis to learn value distribution. RiskQ shares the same converge property as IQN.
>
> As stated in Chapter 7.5 and 7.6 of [3] and Theorem 4.3 of [4], the greedy distributional Bellman update operator of IQN is not a contraction mapping. This is an inherent drawback of the distributional RL. However, the approximation error of some risk metrics can be made arbitrary small by increasing the number of statistics (Theorem 4.7 [4]) for the policy evaluation case.
>
> Recently, Lim and Malik [5] propose a modification to IQN with a new distributional Bellman operator. They show that the optimal CVaR policy corresponds to a fixed point of the new operator. However, it is unclear whether it converges in general settings. We have combined RiskQ with [5] by replacing IQN with it. As shown in Fig. 3 of the response PDF, the new method, LimAndMalki, performs poorly. This suggests that remedying the non-contraction mapping issues may not be important enough for performance improvement as existing risk-sensitive RL methods (e.g., IQN) are already working well.
>
> Further, to verify whether RiskQ optimize toward the risk-sensitive ($Wang_{0.75}$) objective, we show the $Wang_{0.75}$ of the test return distribution for SMAC in Fig. 4 of the response PDF. The results indicate that RiskQ is optimizing the risk-sensitive objective and it performs better than others for that objective.
>
>
> ***likelihood quantile networks ... exploration***
>
> Likelihood Quantile Networks (LQN) [6] assigns various learning rates for samples (from agent exploration or suboptimality) based on likelihoods, which measures return distribution differences. For exploration, LQN gradually anneals the parameters of its risk-based action selection strategy with time.
>
> We have combined RiskQ with the LQN riks-parameters annealing procedure, and call it as RiskQ+LQN. As is depicted in Fig. 5 of the response PDF, RiskQ+LQN performs slightly weaker than RiskQ. This suggests that besides considering the uncertainty for exploration, coordinated exploration may be needed as well.
>
> **same limitations**
>
> Please refer to the global repsonse.
>
> **Reference**
>
> Please refer to the global repsonse.

---

> > ### Comment · Reviewer_e297 · 2023-08-19
> >
> > I appreciate the response and clarifications made by the authors. My questions are mostly answered. Now I do not see any critical downside in this work, I'm increasing my score.

---

> > > ### Author Response · Authors · 2023-08-19
> > >
> > > Thank you for taking the time to review our rebuttal. We are glad to hear that the majority of your concerns have been resolved and there are no significant drawbacks. We are grateful for the increase in the rating.

---

### Author Rebuttal · Authors · 2023-08-06

We faithfully thank all reviewers for their insightful comments and valuable feedback. The reviewers acknowledge our work as an extension of popular principles (X7fQ, rLaN, EN1R), an intuitively sound criterion (e297,EN1R), a good contribution (rLaN, EN1R), novel (EN1R), promising (X7fQ),  with good writing quality (e297, rLaN, EN1R), with extensive theoretical results (e297, X7fQ, rLaN) and thorough experiments (e297, X7fQ, rLaN, EN1R). We will incorporate the suggestions and address the concerns in the new version of this work. We have conducted 17 more experiments to address the comments, and their results are included in the response PDF. We describe new experiments in the response PDF and discuss our limitations.

**Experimental Results**

Fig. 1 (a-c): Results on random SMAC, where one agent performs randomly 50% of the time during testing. These environments are risk-sensitive, and agents should consider cooperation risk, among others.

Fig. 1 (d-f): Results on SMACv2, which is a new version of SMAC with increased uncertainty. Fig. 1 shows that RiskQ performs significantly better than others in risk-sensitive environments.

Fig. 2: An illustrative example of Non-RIGM case and results. Without RIGM, each agent may choose to optimize its risk-sensitive objective separately, which may impede cooperation. Not satisfying the RIGM principle can lead to a significant performance drop.

Fig. 3: Results of replacing the IQN [2] used by RiskQ with a distributional learning algorithm [5] which has better convergence than IQN. It shows that fixing the non-contraction mapping issues of IQN is not that important for a good performance.

Fig. 4: Results of the Wang$_{0.75}$ metric of the test-return and the test-win rate of RiskQ and other algorithms. As an inherent drawback of distributional RL [3-4], IQN used by RiskQ does not guarantee convergence toward the true value distribution, we empirically study whether RiskQ improves the risk-sensitive objective it optimizes during training. The metrics being optimized increase with time, and RiskQ has the best performance.

Fig. 5: Combing RiskQ with the idea of the likelihood quantile network[6]. The risk parameter used by RiskQ gradually anneals with time. This can encourage exploration at the beginning of training. However, we find that using this way alone is not enough for better performance.

Fig. 6: Learning a risk-averse central critic while each agent acts risk-neutrally.

**Limitations**

As we had stated in the submission, RiskQ suffers from representation limitations among quantiles. However, such limitations do not impact the performance signficantly. In RiskQ, the $\omega$-quantile $\theta(\boldsymbol{\tau}, \boldsymbol{u},\omega)$ of the jointed return distribution $Z_{jt}(\boldsymbol{\tau}, \boldsymbol{u})$ is modeled as  $\theta(\boldsymbol{\tau}, \boldsymbol{u},\omega) = \sum_{i=1}^Nk_i\theta_i(\tau_i, u_i, \omega)$, where $\theta_i(\tau_i, u_i, \omega)$ is the $\omega$-quantile of the return distribution $Z_i(\tau_i, u_i)$ of agent $i$, $k_i$ is modeled using the attention mechanism. In this way, RiskQ only models the positive linear relationship among quantile values, and it cannot model functional relationships such as non-monotonic relationships among quantile values. In the submitted paper (Figure 4(e) (f), and Appendix C.5.2), we had explored several RiskQ variants (RiskQ-QMIX, RiskQ-Residual, and RiskQ-Residual-QMIX) that can model more complex functional relationships than RiskQ, and two methods (RiskQ-Residual and RiskQ-Residual-QMIX) are free from such representation limitations. However, in the SMAC environments, RiskQ performs better than its variants which can model more complex relationships.  This suggests that the representation limitations do not impact the performance of RiskQ significantly.

**Same as other value factorization methods**, in RiskQ, obtaining the optimal action over state-action value distribution is intractable.  To derive a practical algorithm, following typical value factorization approaches such as WQMIX, we use approximations to obtain the optimal joint action. That is, during training, the optimal joint action is approximated by $\bar{\mathbf{u}}= [\bar{u_i}]_{i=1}^N$.

$\bar{u_i}= argmax_{u_i}\psi_{\alpha}[Z_i(\tau_i, u_i)]$, $\psi_{\alpha}$ is the risk-metric, $Z_i$ is the individual stochastic utility of each agent. RiskQ may fail to find the correct optimal joint action through this approximation for environments that requires complex coordination. In such scenarios, RiskQ may perform poorly. We believe that combining MARL exploration methods, such as likelihood quantile network [6] and MAVEN [7], could make RiskQ more robust.


**Same as other value factorization methods**, RiskQ is centralized trained with decentralized execution, thus it may suffer from scalability issues. If the number of agents is large, the central training server may fail to allocate enough computational power for the algorithms. Distributed training methods or independent learner algorithms could be used to alleviate this issue. In this work, agents must collaborate to achieve a common goal. Each agent receives a shared rather than individual reward. Thus, RiskQ is not suitable for MARL problems where rewards are mixed.

**References**

[1] Ellis et al. SMACv2: An Improved Benchmark for Cooperative Multi-Agent Reinforcement Learning. arXiv 2022

[2] Dabney et al. Implicit quantile networks for distributional reinforcement learning. ICML 2018

[3] Bellemare et al. Distributional Reinforcement Learning, MIT press, 2023

[4] Rowland et al. Statistics and Samples in Distributional Reinforcement Learning. ICML 2019

[5] Lim and Malik. Distributional reinforcement learning for risk-sensitive policies. NeurIPS 2022

[6] Lyu and Christopher. Likelihood Quantile Networks for Coordinating Multi-Agent Reinforcement Learning, AAMAS 2020

[7] Mahajan et al. MAVEN: multiagent variational exploration. NeurIPS 2019

---

### Decision · Program_Chairs · 2023-09-21

**Decision:**

Accept (poster)

**Comment:**

This paper introduces RiskQ, a risk-sensitive value factorization approach for MARL. The paper formalizes a  Risk-sensitive Individual-Global-Max (RIGM) principle and develops a distributional method (RiskQ) that satisfies it.

While there are some concerns about the strength of the empirical results, the theoretical foundation of the approach is appreciated as well as the practical approach. The method performs well on a number of domains and the additional results provided during the discussion period were helpful.

The responses and resulting discussion were very much appreciated. The authors should make sure to update the paper as described in the discussion, such as the discussion and comparison with other state-of-the-art methods (e.g., LQN), improved clarity and presentation issues (e.g., as pointed out by Reviewer X7FQ), and discussing limitations of the method in more detail.